# OPT-Engine: Benchmarking the Limits of LLMs in Optimization Modeling via Complexity Scaling

Yitian Chen [1]   Cheng Cheng [2 1]   Yinan Sun [2 1]   Zi Ling [3]   Dongdong Ge [4 5]

## Abstract

We investigate the capabilities and scalability of Large Language Models (LLMs) in optimization modeling, a domain requiring structured reasoning and precise formulation. To this end, we introduce OPT-ENGINE, an extensible benchmark framework with quantifiable and controllable complexity. OPT-ENGINE spans ten canonical Operations Research problems, systematically scaling from Linear Programming to Mixed-Integer Programming, providing a structured environment to probe the limits of automated problem formulation and solving. Utilizing OPT-Engine, we address three pivotal research questions. First, we examine whether Pure-Text Reasoning (PTR) via classical Chain-of-Thought can efficiently tackle optimization tasks, finding that PTR suffers from a critical robustness gap as task complexity increases. Second, we examine whether integrating external computational tools can mitigate PTR's arithmetic weaknesses and improve performance. Our results show that while such tools aid local calculations, they still fail to adhere to global optimization constraints. Finally, we pinpoint that for the current SOTA paradigm, Solver-integrated Reasoning (SIR), the automated formulation of constraints represents the primary bottleneck. These findings clarify current paradigms' limitations and provide a structured roadmap for developing next-generation LLMs for optimization modeling. We release our code and data to facilitate future research (https://github.com/Cardinal-Operations/OPTEngine).

[1] Cardinal Operations, Shanghai, China [2] Shanghai University of Finance and Economics, Shanghai, China [3] Booth School of Business, University of Chicago, Chicago, IL, USA [4] Antai College of Economics and Management, Shanghai Jiao Tong University, Shanghai, China [5] Shanghai Institute for Mathematics and Interdisciplinary Sciences, Shanghai, China. Correspondence to: Zi Ling <zling@chicagobooth.edu>.

*Proceedings of the 43rd International Conference on Machine Learning*, Seoul, South Korea. PMLR 306, 2026. Copyright 2026 by the author(s).

## 1. Introduction

Recent advances in frontier large language models (LLMs) (Achiam et al., 2023; Team et al., 2024; Liu et al., 2024; Guo et al., 2025) have shown promising progress on optimization problems modeling and solving. By automatically interpreting natural language descriptions into precise mathematical models, and subsequently generating feasible solutions, current state-of-the-art (SOTA) LLMs significantly lower the barrier to entry for complex optimization techniques, facilitating their accessibility to a broader range of non-experts and accelerating the field's democratization (Antoniou & Lu, 2007; Luenberger et al., 1984).

In parallel, specialized benchmarks (Lu et al., 2025; Huang et al., 2025a; Yang et al., 2025b; Jiang et al., 2025) have emerged to systematically assess LLMs' performances in optimization modeling. By providing curated problem instances paired with canonical solutions derived from expert curation (Huang et al., 2025a) or semi-synthetic pipelines (Lu et al., 2025), these benchmarks serve as a rigorous baseline for evaluation. However, they remain largely confined to static evaluation of aggregate accuracy on fixed-scale instances and are dominated by textbook-style problems that fail to capture real-world complexity. For instance, network-flow tasks in the challenging OptMATH benchmark typically feature instances with around five nodes, which is insufficient to stress high-dimensional constraints and combinatorial challenges inherent in real-world applications. In contrast, the symbolic planning domain has pioneered extensible frameworks like PlanBench (Valmeekam et al., 2023) and Zebralogic (Lin et al., 2025), which establish a paradigm for scaling problem complexity via procedural world-state transitions. Since optimization modeling focuses on the optimality of solutions within a rigorously defined space of variables and constraints, it requires an evaluation paradigm that is distinct from the feasibility focus of planning. An extensible benchmark built on this premise is essential to probe the robustness of LLMs when faced with complexity that vastly exceeds existing benchmarks.

The need for this shift is further underscored by a fundamental methodological bifurcation in how LLMs approach optimization tasks. The current SOTA paradigm, Solver-integrated Reasoning (SIR), focuses on automatic formula-

tion: LLMs first translate natural language descriptions into precise mathematical models (variables, objectives, constraints) and then interface with external solvers like Gurobi (Gurobi Optimization, LLC, 2024) or COPT (Ge et al., 2022) to enhance their computational capacity. This line of work includes multi-agent frameworks (Ramamonjison et al., 2022; AhmadiTeshnizi et al., 2024) and solver-specialized fine-tuning approaches (Huang et al., 2025a; Chen et al., 2025). On the other hand, the recent success of reasoning models, such as OpenAI's o1 (OpenAI, 2024) and DeepSeek-R1 (Guo et al., 2025), has revitalized Purely-text Reasoning (PTR) paradigms. By employing Chain-of-Thought (CoT) approaches (Wei et al., 2022), these models tackle optimization problems through end-to-end deduction (Yang et al., 2024; Jiang et al., 2025). While both report promising results on existing benchmarks (a detailed comparison with current benchmarks is provided in Appendix C.2.2 Table 5), this divergence raises critical questions: Which paradigm is more effective and robust for real-world Operations Research applications? What are the inherent mechanistic differences between these two approaches? A clear and mechanistic understanding of their differences is essential to advance the field beyond isolated demonstrations and side-by-side comparisons.

Furthermore, a stark performance gap persists. On one hand, LLMs have achieved super-human proficiency in competitive mathematics, with even small-scale 4B models (Yang et al., 2025a) surpassing the 95% threshold on the MATH benchmark (Hendrycks et al., 2021), which consists of college-level competition math problems; On the other hand, their application to optimization remains brittle, achieving less than 50% accuracy on the most challenging benchmark (Guo et al., 2023; Chen et al., 2025). The IndustryOR benchmark (Huang et al., 2025a) exemplifies this pattern: it curates realistic tasks with heterogeneous constraints, and reported accuracy remains substantially low even though their resulting mathematical formulations are relatively straightforward. This discrepancy poses another critical question for the community: why persist such a stark performance gap between a model's deductive reasoning capabilities for mathematical reasoning tasks and its capacity for robust optimization modeling?

These observations lead to a set of research questions that the current existing benchmark cannot adequately answer. To fill this gap, we present OPT-ENGINE, an extensible benchmark framework designed to move beyond static and anecdotal evaluations and to provide a rigorous assessment of LLMs' capacity for auto-formulation and problem solving. OPT-ENGINE programmatically generates a vast repository of problem instances with controllable structure and difficulty, flexible natural language specifications, and verifiable optimal solutions. This design enables a new evaluation paradigm and provides the empirical evidence necessary to advance the field toward industrial-scale optimization modeling

In summary, our contributions are fourfold: 1.) We introduce OPT-Engine, a comprehensive framework that enables extensible benchmarking of LLMs in optimization modeling with fine-grained, scalable control of mathematical structure and language. 2.) We evaluate PTR via classical Chain-of-Thought and find that PTR suffers from a critical robustness gap as task complexity increases. In contrast, integration of an external solver proves crucial for maintaining accuracy at scale. 3.) We investigate whether integrating external computational tools can mitigate PTR's arithmetic weaknesses. Our results show that while such tools help with local calculations, they still fail to adhere to global optimization constraints. 4.) Through systematic experimentation, we pinpoint that for the SOTA, Solver-Integrated Reasoning paradigm, the automated formulation of constraints, rather than problem comprehension or objective perturbations, is the primary performance bottleneck.

## 2. Related Work

**Benchmarks for Optimization Modeling.** The development of LLMs for optimization modeling has been accelerated by the creation of specialized benchmarks. This line of work was notably advanced by the LP word problem dataset and shared optimization tasks for the NeurIPS 2022 competition (Ramamonjison et al., 2023). Subsequent benchmarks like MAMO (Huang et al., 2025c), IndustryOR (Huang et al., 2025a), OptMATH (Lu et al., 2025), and OPTIBENCH (Yang et al., 2025b) have expanded the scope from linear to nonlinear and mixed-integer programming. The second stream, including ALE-Bench (Imajuku et al., 2025) and NP-engine (Li et al., 2025), focuses on heuristic solutions to combinatorial problems but does not provide exact solutions for evaluations. Furthermore, while frameworks like OptMATH utilize programmatic pipelines for problem generation, they prioritize training data synthesis over the high-fidelity solution accuracy necessary for robust benchmarking.

**Paradigms in LLM-driven Optimization.** Current research on LLMs for optimization broadly falls into two categories. The first is Solver-integrated reasoning (SIR), which follows an "Autoformulation-then-Solve" (Huang et al., 2025a; Chen et al., 2025; Nguyen & Ko) paradigm, utilizing LLMs as formulation and code generators to interface with external optimization solvers such as Gurobi or COPT. These tool-augmented methods leverage code data to enhance reasoning, effectively offloading heavy computation to external solvers. In contrast, Pure-text reasoning (PTR) approaches (Yang et al., 2024), exemplified by recent advanced reasoning models (OpenAI, 2024; Guo et al., 2025), seek to solve optimization problems within the tex-

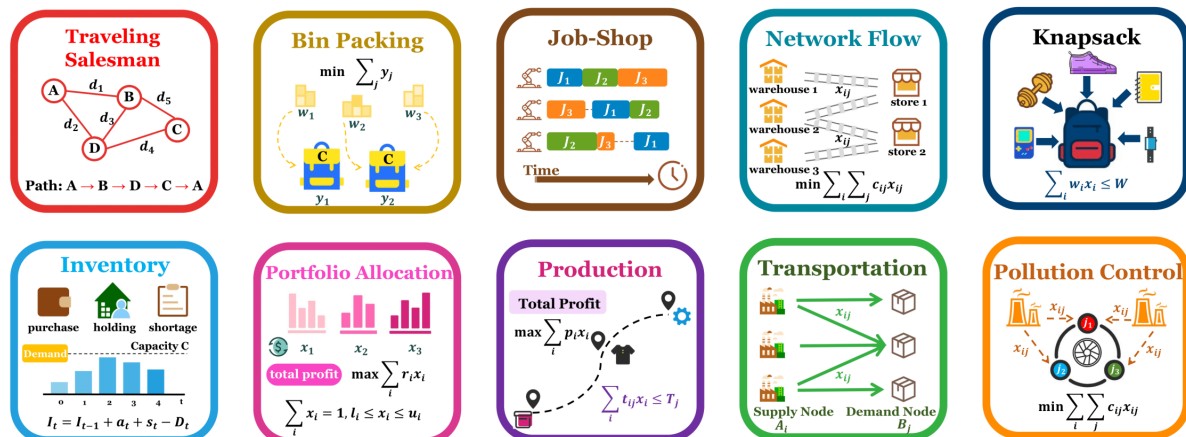

*Figure 1.* An overview of the OPT-Engine taxonomy. The framework categorizes ten problem classes into two tiers: five Mixed-Integer Programming (MIP) problem classes in the top row and five Linear Programming (LP) problem classes in the bottom row.

tual domain through prompt-based methods (Yang et al., 2024; Tang et al., 2025) or supervised training (Jiang et al., 2025). Despite the promise of these dual trajectories, a rigorous empirical analysis is still required to characterize their respective strengths and limits across scales of problem complexity.

**Benchmarks with Scalable Complexity and Structural Perturbations.** Prior work in symbolic logic and planning (Lin et al., 2025; Valmeekam et al., 2023; Kokel et al., 2025) has established the paradigm of scaling problem complexity through world-state transitions. Based on these benchmark frameworks, a series of systematic evaluations of frontier LLMs have been conducted (Song et al., 2025; Valmeekam et al., 2024; 2025). These studies reveal clear strengths and limitations in LLMs' reasoning, particularly as problem complexity increases, and demonstrate phenomena such as "reasoning illusions" where model performance degrades significantly as scale grows (Shojaee et al., 2025). While prior work mainly identifies LLMs' sensitivity to linguistic shifts (Mirzadeh et al., 2025; Hong et al., 2025) and structural perturbations (Huang et al., 2025b) in general mathematics, we transpose these diagnostics to the optimization domain. We systematize this evaluation through a tripartite decomposition: linguistic variance, objective perturbations, and constraint disturbances.

## 3. OPT-Engine: Taxonomy and Pipeline Framework

### 3.1. OPT-Engine: Benchmark Taxonomy.

As shown in Figure 1, the OPT-Engine framework encompasses ten canonical problem classes that cover the breadth of real-world optimization categories essential to Operations Research (OR) applications. Specifically, the frame-

work is structured into two primary categories: 1.) **Linear programming (LP).** This family comprises five optimization problem classes, including the inventory problem, the portfolio allocation problem, the production problem, the transportation problem, and the pollution control problem. 2.) **Mixed-integer programming (MIP).** This category includes five combinatorial optimization classes with integer or binary constraints: the traveling salesman problem (TSP), the knapsack problem, the bin packing problem, the job-shop scheduling problem and the minimum-cost network flow problem.

Crucially, the framework is designed to generate problem instances with scalable and controllable complexity. For each problem class, the difficulty of a generated problem instance is modulated by tuning specific structural parameters. For example, the difficulty of the TSP can scale with the number of cities $N_{\text{cities}}$, while the portfolio allocation problems can scale with the number of assets $N_{\text{assets}}$. This design enables fine-grained and systematic evaluation across difficulty levels, thereby facilitating rigorous tests of LLMs' ability to generalize across the full range of problem scale and constraint density.

We next detail the pipeline workflow, illustrating how OPT-Engine synergistically combines programmatic instance generators, natural language problem construction, and LLM-powered augmentation to achieve controlled generation and verified solutions.

### 3.2. OPT-Engine: Pipeline Framework

Given an optimization problem class $d \in \mathcal{D}$, where $\mathcal{D} = \{d_1, d_2, \ldots, d_n\}$ denotes the ten problem classes introduced above, we detail the pipeline that produces fully specified, verifiable instances from each class. As shown in Figure 2, the overall pipeline, which generates the final problem in-

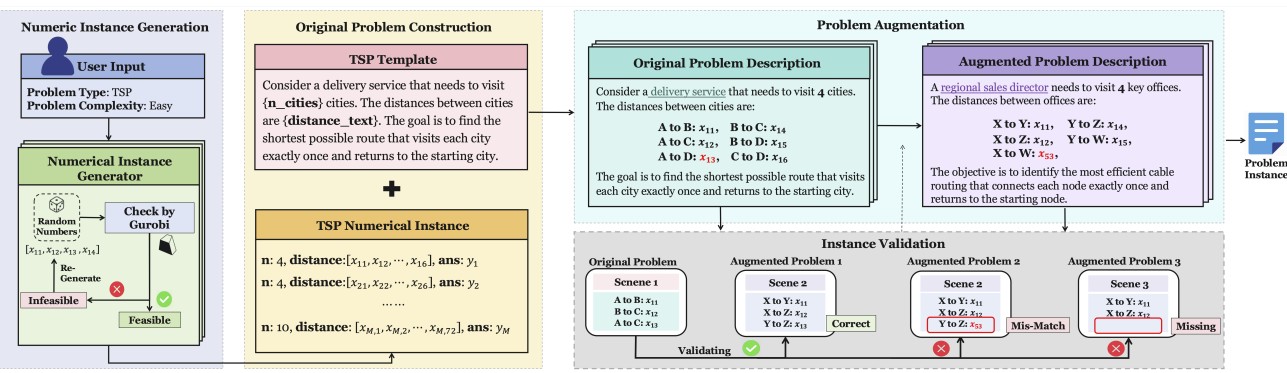

*Figure 2.* The OPT-ENGINE workflow. The four-stage pipeline for problem instance generation: (1) numeric instance generation, (2) original problem construction, (3) problem augmentation, and (4) instance validation.

stance (including its type, complexity, problem description, and verifiable solution), can be divided into four main stages: 1.) the **numeric instance generation** stage, 2.) the **original problem construction** stage, 3.) the **problem augmentation** stage, and 4.) the **instance validation** stage.

In the first stage, the numeric instance generator $G$ takes the problem class $d$ and a difficulty control parameter vector $\theta$ as the input, and generates the core numeric data that represents a specific numeric instance. This process can be formally viewed as a function $G$ that maps the problem class $\mathcal{D}$ and control space $\Theta$ to the numeric instance space $\mathcal{I}$: $G : \mathcal{D} \times \Theta \to \mathcal{I}$.

Here, $\Theta \subseteq \mathbb{R}^k$ represents the $k$-dimensional space of controllable difficulty parameters (e.g., the number of cities for the TSP, where $k = 1$). Crucially, $G$ is coupled with a solver template specific to class $d$. This code template is essential for generating the exact optimal objective value and decision variables that serve as the ground truth for subsequent validation stages. For example, a size parameter $n$ induces a controllable $n \times n$ distance matrix for TSP, after which the solver computes the optimal solution. Meanwhile, $G$ is equipped with a self-correction mechanism: if a draw is infeasible, the generator resamples until a valid numeric instance is produced.

Once a valid numeric instance $i \in \mathcal{I}$ is available, the pipeline proceeds to the "canonical problem construction" stage. In this stage, we utilize a structured problem description template $T$ to map the instance $i$ into the canonical problem statement $s_c \in \mathcal{S}_C$. This generated statement serves as the essential, formal original optimization problem used for subsequent rephrasing and validation. This mapping $M$ is defined as: $M : \mathcal{I} \times T \to \mathcal{S}_C$.

In the subsequent "problem augmentation" (or rephrasing) stage $R$, a dedicated LLM agent $\mathcal{L}$, is employed to diversify the problem descriptions. Specifically, $\mathcal{L}$ ingests the canonical problem statement $s_c$ and leverages the LLM's gener-

ative capability to produce a set of context-rich, domain-specific narratives $s_r \in \mathcal{S}_R$, which render the original problem into multiple simulated Operations Research scenarios. This transformation is defined as: $R : \mathcal{S}_C \times \mathcal{L} \to \mathcal{S}_R$.

The final phase of the pipeline is "problem instance validation". This is carried out by a validation module ($\mathcal{V}$) that combines an LLM as a Judge (Zheng et al., 2023) with a rule-based verifier. By simultaneously checking for numerical correctness and structural preservation, this module guarantees the consistency of the constraints and objectives across the original canonical problem $s_c$ and the generated rephrased statements $s_r$.

If an augmented problem fails validation, the system repeats $R$ until a valid problem instance is obtained. This recursive loop ensures that the final output, comprising the problem type, complexity metrics, rephrased narrative, and verifiable solution, maintains an analytically optimal value consistent with the ground truth derived from the numeric generator $G$. The full implementation details, including detailed class definitions with corresponding templates and required prompt templates during the pipeline, are provided in Appendix B.

## 4. SIR vs. PTR Approaches

We first compare and evaluate the relative efficacy of SIR and PTR on the benchmark dataset generated by the proposed OPT-Engine framework, examine how their performance and reasoning patterns evolve with increasing problem dimensionality and combinatorial complexity. We then probe deeper: can addressing the computational drawback of PTR enable it match the performance of SIR?

### 4.1. Definitions and Experimental Setup

Let $\mathcal{X} = \{x_i\}_{i=1}^N$ denote the evaluation set of $N$ optimization problem instances generated by OPT-Engine. For each problem $x_i \in \mathcal{X}$, we prompt the LLM to generate a sequence of intermediate reasoning steps $\mathbf{z}^{(i)} =$

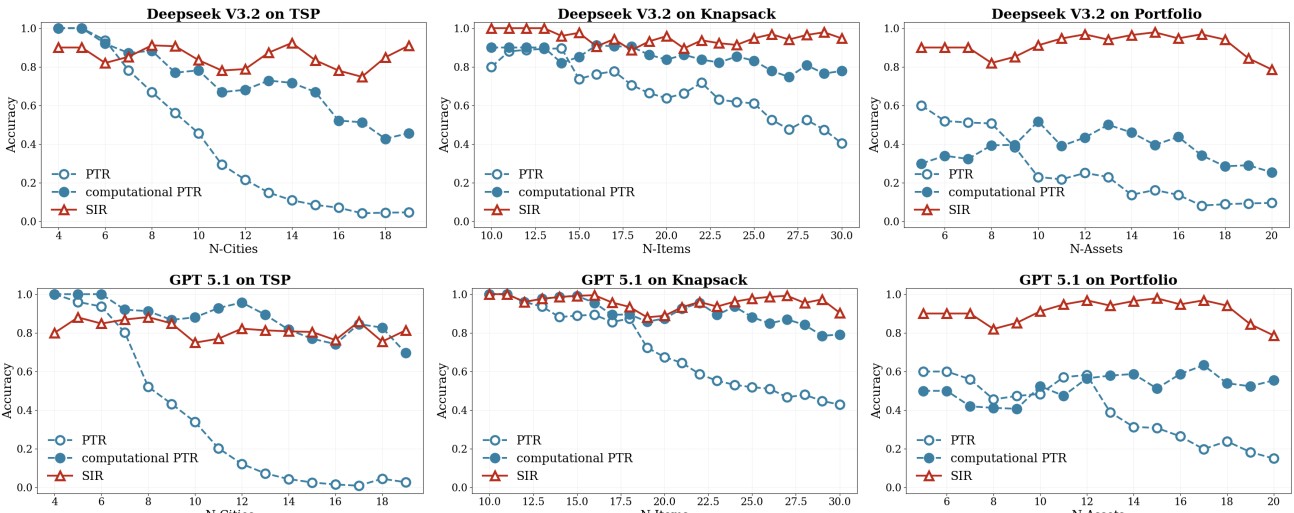

*Figure 3.* Performance scaling of PTR (blue, dotted), computational PTR (blue, solid), and SIR (red, dashed) on frontier models. The upper panel shows results for the DeepSeek V3.2 model, and the lower panel shows results for the GPT-5.1 model.

$(z_1^{(i)}, z_2^{(i)}, \ldots, z_m^{(i)})$ with terminal step $z_m^{(i)}$. We then run two approaches $p \in \{\text{PTR}, \text{SIR}\}$ that differ in how they process this terminal step $z_m^{(i)}$ to derive the objective value $\hat{y}_i^{(p)}$. In PTR, the LLM is prompted to reason sequentially to derive the optimal objective value $\hat{y}_i^{(\text{PTR})}$, which is contained within the final reasoning step $z_m^{(i)}$. In SIR, the terminal step $z_m^{(i)}$ contains an executable code snippet, which we run in an external execution solver to obtain $\hat{y}_i^{(\text{SIR})}$. The corresponding prompt templates are provided in the Appendix C.1.1.

To evaluate robustness, we generate ten distinct instances for each problem class at each complexity level. A solution $\hat{y}_i^{(p)}$ is considered correct if the relative error compared to the ground-truth optimum $y_i^*$ satisfies:

$$\frac{\left| \hat{y}_i^{(p)} - y_i^* \right|}{|y_i^*| + 10^{-6}} < 10^{-3}.$$

For each complexity level, we report `avg@10`, representing the mean success rate across these ten instances, and it serves as the standard accuracy measure throughout the remainder of the paper.

### 4.2. Comparative Analysis: SIR vs. PTR

**Comparative Analysis with Top-Tier Models.** In the first phase of the comparative study, we utilized two proprietary API-Accessed LLMs: DeepSeek-V3.2 (Liu et al., 2025) and GPT-5.1 (Singh et al., 2025). By leveraging these top-tier models, we aim to rigorously evaluate the performance of these two approaches, thereby establishing a strong baseline that reflects these approaches' intrinsic capabilities. As illustrated in Figure 3, performance trends are consistent across all problem classes, including both LP and MILP. As

the complexity of the problem increases, SIR sustains high accuracy or exhibits only minor degradation. In contrast, the performance of PTR degrades substantially with scale.

**Comparative Analysis with Weaker Models.** We then extended our analysis to Qwen3-4B-Instruct (Yang et al., 2025a). This model's strong instruction-following performance makes it an ideal candidate for examining whether the observed reasoning advantages persist at a smaller model scale.

As highlighted by the dashed lines in Figure 4, PTR outperforms SIR at low-complexity settings. This observation contrasts with the scaling trends in our analysis of frontier models. However, this advantage is transient: the efficacy of PTR significantly degrades as the problem scale expands. We attribute this performance gap to the limited code-generation capabilities of smaller models under the SIR approach, as evidenced by the low execution rates in Table 1. By addressing this core limitation through RLVR post-training (Chen et al., 2025) (see Appendix C.1.2 for details), we obtain Qwen3-4B-RL. As shown in Figure 4, Qwen3-4B-RL, initialized from Qwen3-4B-Instruct and further optimized , consistently outperforms the standard PTR approach across all problem dimensions.

*Table 1.* Execution Rate Comparison Across Models

| Model | TSP | Knapsack | Inventory |
|---|---|---|---|
| Deepseek-V3.2 | 86.4% | 100.0% | 88.8% |
| GPT-5.1 | 83.1% | 99.6% | 92.5% |
| Qwen3-4B-Instruct | 23.1% | 23.1% | 35.6% |
| Qwen3-4B-RL | 38.1% | 99.6% | 38.1% |
| $\Delta$ | +15.0% | +76.5% | +2.5% |

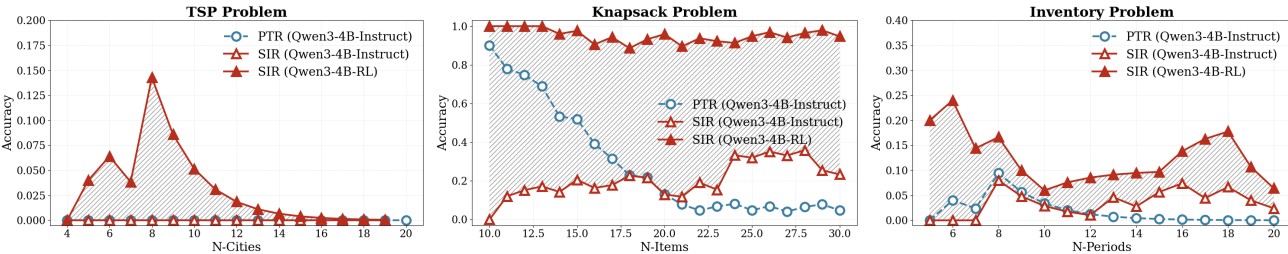

*Figure 4.* **Performance scaling of PTR (blue) vs. SIR (red) on the Qwen3-4B series**. The panel displays results from Qwen3-4B-RL, indicating RLVR training significantly enhances SIR mode accuracy.

**Comparative Analysis: A Synthesis of Critical Findings.** Our experimental results support two conclusions. First, when using PTR without external computational tools , LLMs hit a severe performance ceiling as problem complexity scales. This decline reflects a core mismatch between LLMs' intrinsic capabilities of textual reasoning and the exact algorithmic computations required in formal optimization, e.g., the $O(n^3)$ operations of the Simplex method for LP problems (Luenberger et al., 1984). Second, by generating executable solver codes and invoking tool-calling interfaces, LLMs consistently outperform their tool-free counterparts. This delegation effectively offloads the exact computational burden to deterministic engines, maintaining robustness where PTR fails. Consequently, our findings establish SIR as the requisite framework for addressing high-complexity, industrial-scale optimization challenges.

### 4.3. Beyond Arithmetic: Computational Aid vs. Structural Limits

So far, we have demonstrated that PTR performance degrades with increasing problem size, a stark contrast to earlier reports of PTR success (Jiang et al., 2025; Tang et al., 2025), the underlying failure modes necessitate a more granular investigation. To gain deeper insight into the behavior of PTR and obtain mechanistic understanding, we design and conduct a three-stage systematic analysis. We first analyze token-level response patterns and conduct detailed case studies of full reasoning traces. Next, we implement an error-decomposition experiment that distinguishes between **constraint infeasibility** and other errors, including **computational inaccuracy**. By defining infeasibility as a violation of the feasible region, we effectively decouple structural modeling failures from arithmetic noise for PTR. Finally, we investigate whether external computational tools facilitate global feasibility or merely resolve local arithmetic weaknesses while leaving the underlying constraint logic unresolved.

**Case Study: Failure Signatures in PTR Responses.** Figure 5 illustrates the relationship between average output length and accuracy for TSP under DeepSeek-V3.2. At small scales, the model increases its apparent reasoning ef-

fort, measured by output tokens count, roughly in proportion to problem complexity. However, as the city size approaches a critical range, closely preceding the point of accuracy collapse, the model counterintuitively reduces its token budget despite increasing task difficulty. Moreover, we provide a case study in Figure 13 which clarifies the strategies behind this shift in Appendix C.2.3. In the initial phase with small problem sizes, the model predominantly relies on explicit enumeration of Hamiltonian cycles to solve the problem exactly, because the search space remains tractable (size $(n-1)!$). This yields high accuracy with growing token consumption in this phase. However, beyond the critical threshold, the model shifts to employing lightweight heuristics such as nearest neighbor and cheapest insertion to obtain approximate solutions, which holds token usage approximately flat but consequently lowers solution quality.

The resulting PTR behavior is adaptive in a way that resembles expert practice, opting for fast heuristics when exact solutions become computationally prohibitive. Furthermore, this dynamic provides a mechanistic explanation for the efficacy of PTR-based end-to-end optimizers reported in prior studies (Jiang et al., 2025; Tang et al., 2025). Their efficacy in combinatorial optimization stems from an inherent ability to dynamically shift to heuristic methods and evaluation metrics emphasizing near-optimal, practical solutions over strict optimality.

**Constraint Feasibility vs. Computational Accuracy.** The previous case study suggests that PTR behaves as a "stochastic approximation", producing directionally correct yet nu-

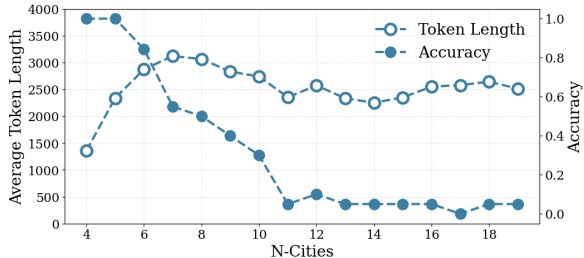

*Figure 5.* **Scaling behavior on TSP**: relationship between token length and accuracy across increasing problem complexities.

| Problem | Approaches | Easy | | Medium | | Hard | |
|---|---|---|---|---|---|---|---|
| | | Feas/Acc | OptGap | Feas/Acc | OptGap | Feas/Acc | OptGap |
| **TSP** | PTR | 73.3% / 58.3% | 1.07% | 70.0% / 10.0% | 10.14% | 64.0% / 2.0% | 11.72% |
| | CompPTR | 86.7% / 85.0% | 0.15% | 72.0% / 70.0% | 0.05% | 52.0% / 44.0% | 0.43% |
| | SIR | 98.3% / 91.7% | 0.01% | 96.0% / 86.0% | 0.25% | 88.0% / 80.0% | 0.24% |
| **Knapsack** | PTR | 93.9% / 89.2% | 5.62% | 93.2% / 78.0% | 0.29% | 61.8% / 45.6% | 0.42% |
| | CompPTR | 98.5% / 87.7% | 0.00% | 98.3% / 83.3% | 0.01% | 87.1% / 78.6% | 0.00% |
| | SIR | 100.0% / 92.3% | 0.00% | 100.0% / 91.7% | 0.00% | 100.0% / 95.7% | 0.00% |
| **Netflow** | PTR | 40.0% / 10.0% | 3.57% | 15.0% / 0.0% | 2.64% | 0.0% / 0.0% | 0.00% |
| | CompPTR | 90.0% / 90.0% | 0.00% | 85.0% / 85.0% | 0.00% | 75.0% / 75.0% | 0.00% |
| | SIR | 100.0% / 100.0% | 0.00% | 100.0% / 100.0% | 0.00% | 100.0% / 100.0% | 0.00% |

*Table 2.* Performance comparison across problem classes (from Easy to Hard): feasibility/accuracy rates and optimality gaps for PTR, CompPTR, and SIR approaches on the DeepSeek-V3.2 model.

merically imprecise results. To mechanistically validate this hypothesis, we conduct an error-decomposition experiment that distinguishes between constraint feasibility and computational accuracy. (For the feasibility criteria used for PTR, see Appendix A.2 for details.)

As illustrated in Table 2, we decompose errors into two categories: infeasibility errors (constraint violations) and other errors (e.g., computational inaccuracy). Correspondingly, we report both the feasibility rate and accuracy results. Comparing SIR and PTR through this error decomposition reveals a fundamental trade-off as problem difficulty scales from **Easy** to **Hard**: 1.) In SIR paradigm: accuracy is strictly contingent on structural feasibility. Because numerical execution is offloaded to an external solver, the bottleneck is purely structural. If the model captures the correct logical grounding (i.e., constraints and objectives), the solver guarantees optimality. 2.) In contrast, PTR struggles with both feasibility and computational inaccuracy as complexity increases, exhibiting a dual-failure trajectory: as complexity scales, the model first experiences 'reasoning collapse,' failing to satisfy the global logical constraints necessary for feasibility. Second, even when the model identifies the correct logical manifold, it lacks the deterministic numerical precision required to reach exact mathematical optima."

**Can Computational Tools Aid Pure-Text Reasoning?** A natural question then follows: does augmenting LLMs with computational tools enable PTR to overcome its limitations and lead to performance comparable to SIR? To investigate this, we introduce CompPTR, an extension of PTR that augments natural language reasoning with Python-based computational primitives (e.g., via `scipy` or `itertools`). While the model is permitted to compose these snippets to synthesize a final solution, it is explicitly restricted from invoking external solvers for global optimization.

The results in Table 2 places CompPTR at an intermediate performance level, outperforming PTR but not yet matching SIR. We observe marked improvements in both feasibility and numerical precision, This suggests that external computational support mitigates not only arithmetic errors but also some underlying reasoning failures that previously led to infeasibility. Despite these gains, CompPTR still underperforms relative to SIR due to reasoning collapse as problem complexity increases. While the model successfully leverages computational tools for discrete sub-tasks, it fails to internalize the holistic combinatorial dependencies of the optimization problem. Consequently, CompPTR cannot maintain a coherent global optimization strategy, and its performance degrades as problem size grows.

## 5. The Primary Bottleneck: Diagnosis and Evidence

We now systematically diagnose the primary bottlenecks of current LLMs in optimization. Informed by recent advances in mathematical reasoning diagnostics (Mirzadeh et al., 2025; Hong et al., 2025; Huang et al., 2025b), we examine three primitive factors: 1.) linguistic complexity in problem descriptions, 2.) perturbations to the objective functions, and 3.) augmentation of constraint conditions. The first axis probes the textual surface of the task, whereas the latter two are intrinsically tied to the optimization problem's mathematical structure that drives solving. We present results for the DeepSeek-V3.2 model in the main text; results for GPT-5.1 are provided in Appendix D.1.

### 5.1. Diagnostic Experiments

**Linguistic Complexity.** We first test the causal role of linguistic complexity by constructing a controlled template experiment. From a canonical description template, we create two additional derivative versions with progres-

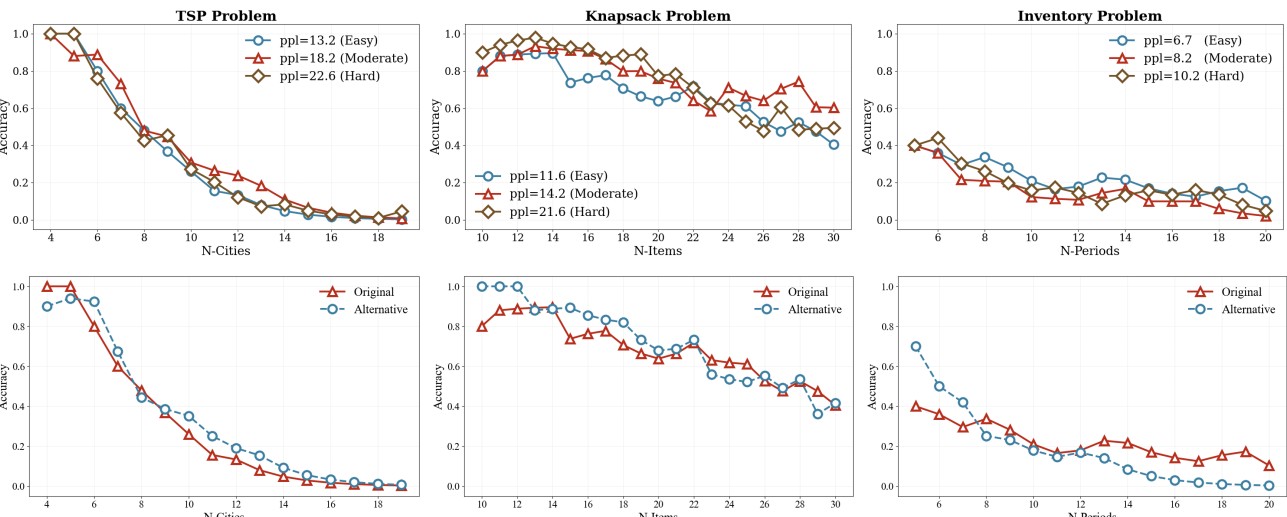

*Figure 6.* **Performance scaling on the Deepseek-V3.2 Model**. **Top Row**: Accuracy under varying linguistic complexity (Easy, Moderate, and Hard tiers, measured by PPL); **Bottom Row**: Accuracy comparison between the original and perturbed objective functions.

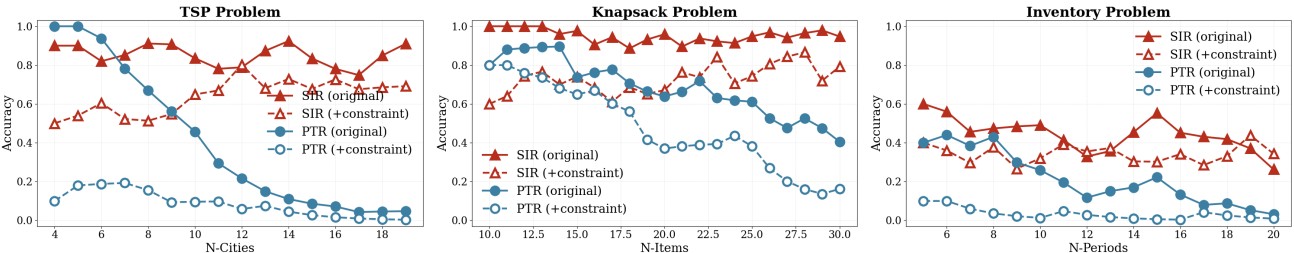

*Figure 7.* **Performance scaling under baseline vs. augmented constraints on the Deepseek-V3.2 Model.** PTR is shown in blue and SIR in red. Augmented constraints used in evaluation are as follows: **TSP**: Exactly one of the following two roads must be included in the tour: the road between city 1 and city 2, the road between city 2 and city 3. ($x_{01} = 0, x_{12} + x_{23} = 1$); **Knapsack**: Exactly one of item 1 and item 2 must be selected. If item 3 is selected, the effective backpack capacity is reduced by 2 kg. ($x_1 + x_2 = 1, \sum_{i=1}^{n} w_i x_i \leq C - 2x_3$); **Inventory**: On day $t$, the on-hand inventory must be at least $I_{\min}$ units. ($I_t \geq I_{\min}$). The introduction of augmented constraints leads to a consistent accuracy drop across problem classes.

sive syntactic and lexical complexity. Critically, all three templates share the *same* numeric instance and constraint set, isolating linguistic variation as the sole independent factor. To rigorously quantify this variation, we employ perplexity(PPL) (Jelinek, 1980) (see Appendix A.3 for details) as our primary complexity metric. Figure 9 exemplifies these variations for TSP across three distinct complexity tiers (easy, moderate and hard).

Building on this metric, we investigate its impact on model performance. The results in the top row of Figure 6 show that solution accuracy remains stable as PPL increases when the underlying mathematical structure is held fixed.

**Objective Perturbations.** We next investigate whether perturbations to the objective function disrupt LLM-based optimization modeling accuracy. Intuitively, modifying the objective function shifts the optimization criterion and could potentially confuse models that rely on familiar signatures rather than explicitly reasoning about objectives. Formally,

for the original objective $f(x)$, we apply a linear perturbation by adding a constant:

$$\min_x f(x) \implies \min_x f(x) + K, \tag{1}$$

where $K$ is a randomly sampled constant that does not depend on the decision variables. The optimal solution remains invariant under this constant shift.

As demonstrated in the bottom row of Figure 6, this perturbation has a negligible impact on accuracy across all problem classes, indicating that such objective shifts also do not function as a key bottleneck for optimization modeling.

**Constraint Augmentation.** To evaluate the impact of augmented constraint descriptions, we construct variants for each problem class by introducing a set of mathematically straightforward constraints. This setup reflects the multi-constraint nature of real-world optimization while maintaining a controlled test environment. By construction, each addition preserves the formal problem class and in-

| TSP | Knapsack | Inventory |
|---|---|---|
| **Augmented constraint description:** There is no direct road between city 0 and city 1. Exactly one of the following two roads must be included in the tour: the road between city 1 and city 2, the road between city 2 and city 3. 

 **Constraint formulation:** $x_{01} = 0,\ x_{12} + x_{23} = 1.$ | **Augmented constraint description:** Exactly one of item 1 and item 2 must be selected. If item 3 is selected, the effective backpack capacity is reduced by 2 kg. 

 **Constraint formulation:** $x_1 + x_2 = 1,\ \sum_{i=1}^{n} w_i x_i \ \leq\ C - 2x_3.$ | **Augmented constraint description:** On day $t_1$, the maximum order quantity is $Q_{\text{cap}}$ units. On day $t_2$, the on-hand inventory must be at least $I_{\text{min}}$ units. 

 **Constraint formulation:** $o_{t_1} \leq Q_{\text{cap}},\ I_{t_2} \geq I_{\text{min}}$ |

*Figure 8.* Augmented constraint descriptions and their corresponding mathematical formulations across different problem types.

stance size: no new variables are introduced and only $\mathcal{O}(1)$ constraints are added, so the intrinsic mathematical difficulty and asymptotic complexity remain unchanged even if the optimal solution may differ. Consequently, the modeling burden shifts from computation to semantics, as the model is required to accurately parse and integrate auxiliary conditions into the formal formulation.

Figure 7 illustrates representative augmentations and their exact formulations: for TSP, we forbid one edge and require exactly one of two others; for Knapsack, we impose mutual exclusivity and a capacity shift; and for Inventory, we add order caps and minimum stock levels. Quantitative results for PTR and SIR indicate significant accuracy degradation relative to canonical problems. This decline is counterintuitive given the comparable mathematical complexity; however, failure traces suggest that formulation errors, such as omitted constraints, propagate to solvers in SIR or yield incorrect objectives in PTR.

### 5.2. Identifying the Primary Bottleneck

Our results show that LLM-based solution accuracy remains largely robust to both increased linguistic complexity and objective-function perturbations when the underlying problem structure is preserved. In contrast, adding even simple constraints leads to a substantial accuracy drop across problem classes under both SIR and PTR paradigms, highlighting augmented constraints as a primary bottleneck for LLM-based optimization modeling.

This counterintuitive pattern indicates that current LLMs often do not fully internalize or reason robustly about problem constraints, but instead rely on patterns tied to familiar, canonical formulations that are likely to be abundant in their training data. The canonical versions of TSP, Knapsack, and related problems closely resemble textbook examples, whereas the constrained variants introduce novel combinations of conditions that the model is less likely to have studied. However, real-world optimization tasks rarely appear as clean canonical problems and typically induce additional constraints tailored to complex operational or regulatory requirements. Our findings, therefore, indicate that OPT-Engine provides a useful framework for probing whether LLMs can move beyond solving stylized textbook instances and handle the richer, constraint-heavy optimization problems that arise in real applications.

## 6. Conclusion and Limitations

By systematically scaling problem complexity, OPT-ENGINE establishes a new protocol for benchmarking LLMs in optimization modeling. Our evaluation identifies Solver-Integrated Reasoning as essential for maintaining reliability as complexity increases. In contrast, Pure-Text Reasoning via Chain-of-Thought exhibits a critical robustness gap under scaling. Moreover, integrating external computational tools mitigates PTR's arithmetic weaknesses and assists with local calculations, but still fails to enforce global optimization constraints. Finally, we expose a critical "semantic sensitivity" bottleneck: even frontier models struggle to maintain formulation fidelity when constraint specifications deviate from canonical problem versions. Therefore, OPT Engine offers the diagnostic rigor and roadmap necessary to build models capable of addressing real-world optimization challenges. While these findings reflect an exact-solution paradigm, we acknowledge that exact modeling is often computationally prohibitive for large-scale NP-hard problems where high-performance heuristics are preferred. As future work, it is worth adding tracks that report optimality gaps, runtime, and robustness under time budgets, and evaluating LLMs' ability to design or select effective heuristics with different training signals and evaluation criteria. Finally, the current benchmark focuses in linear structure in LP and MIP, so extending OPT-ENGINE to nonlinear, stochastic, or dynamic programs remains an important direction.

## Acknowledgements

We thank Siyu Shao of the University of Hong Kong for discussions that helped shape the initial framework of the paper. We also thank the anonymous reviewers for their valuable feedback. This research is partially supported by the National Natural Science Foundation of China (NSFC) [Grant NSFC-72225009, 72394360, 72394365].

## Impact Statement

OPT-ENGINE establishes a rigorous, scalable benchmark for evaluating Large Language Models (LLMs) in the domain of optimization modeling and enables researchers to systematically identify the proficiencies and bottlenecks of AI systems across a spectrum ranging from fundamental problems to high-complexity, approximated real-world OR problems. By utilizing LLM agents to synthesize abstract problems, OPT-ENGINE minimizes reliance on sensitive datasets while maintaining high-fidelity testing. We believe that this framework will accelerate the development of robust, reliable LLM-driven optimization methodologies, fostering a more transparent understanding of their current limitations and future potential.

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

# A. Technical Background

## A.1. Auto-formulation of Optimization Problems

In this work, auto-formulation denotes the task of using an LLM-based agent to transform a human-readable problem description into this formal representation and its executable counterpart. The formal target of auto-formulation is the canonical optimization problem form (Luenberger et al., 1984):

$$
\begin{aligned}
\min_{\mathbf{x}} \quad & g(\mathbf{x}), \\
\text{subject to} \quad & c_i(\mathbf{x}) = 0, \quad i \in \mathcal{E}, \\
& c_i(\mathbf{x}) \geq 0, \quad i \in \mathcal{I},
\end{aligned}
\tag{2}
$$

where $\mathbf{x} \in \mathbb{R}^n$ denotes the decision vector, and the objective function $g : \mathbb{R}^n \to \mathbb{R}$ assigns a scalar value to each candidate solution $\mathbf{x}$. The constraint functions $c_i : \mathbb{R}^n \to \mathbb{R}$ define the feasible region through equality constraints indexed by $\mathcal{E}$ and inequality constraints indexed by $\mathcal{I}$.

This auto-formulation process involves three distinct representational components:

- **Natural Language Problem.** A natural language problem is an unstructured textual description of a decision-making scenario, often presented as a word problem or real-world query (e.g., How should a factory allocate resources to maximize profit given limited labor and materials?). These problems are intuitive for humans but ambiguous and non-executable for computers. It's necessary to translate into formal mathematical representations.

- **Mathematical Model.** This process yields a precise abstraction of the problem, necessitating the explicit extraction and definition of decision variables, the objective function, and mathematical constraints. The resulting model serves as a critical interpretable artifact, effectively bridging the gap between natural language and computational execution

- **Solver Code.** The final executable output consists of code that instantiates the mathematical model for a specific solver, such as Gurobi or COPT. This step translates the abstract formulation into the syntactic format required by numerical optimization engines, effectively bridging the gap between theoretical description and computational solution.

## A.2. Feasibility Rate for PTR and CompPTR

While standard accuracy metrics measure exact convergence to a global optimum, we introduce the Feasibility Rate ($\Phi$) to facilitate an error-decomposition analysis that distinguishes between structural infeasibility and numerical inaccuracy.

**Definition A.1** (Feasibility Rate). For a generated decision variable output $\hat{x}$, we assign a binary feasibility label based on the following criteria:

- Structural dimensionality: The vector $\hat{x}$ must strictly adhere to the required problem dimensionality (e.g., $\hat{x} \in \mathbb{R}^d$), reflecting the model's comprehension of the latent problem space.

- Constraint adherence: The variables must satisfy the full system of constraints provided in the problem manifold (e.g., $A\hat{x} \leq b, \hat{x} \geq 0$).

A solution is labeled feasible if it satisfies both conditions; otherwise, it is deemed infeasible.

The feasibility rate is defined as the proportion of feasible solutions across a given problem cohort, serving as a proxy for the model's structural integrity independent of its final optimality. Analogous to mean accuracy, we report for each complexity level avg@10, the mean feasibility rate across the ten instances.

## A.3. Perplexity: PPL

In NLP domain, PPL serves as an intrinsic measure of how well a probability distribution predicts a sample, effectively capturing the "surprise" encountered by an LLM during parsing. For a given sequence $X$, the formulation is given by:

$$
\text{PPL}(X) = \exp\left\{ -\frac{1}{t} \sum_i^t \log p_\theta(x_i | x_{<i}) \right\}
\tag{3}
$$

where $\log p_\theta(x_i|x_{<i})$ is the log-likelihood of the $i$-th token conditioned on the preceding tokens $x_{<i}$ according to the model $\theta$.

In the context of our benchmark, PPL provides a granular metric for problem statement complexity. A low perplexity score implies that the optimization problem is described using standard, high-frequency terminology and simple syntactic structures. Conversely, a high perplexity score identifies complex problem descriptions, often characterized by specialized jargon, nested constraints, or unconventional phrasing, which present a higher cognitive load for the LLM to parse and formulate.

### A.3.1. EXPERIMENTAL DETAILS OF BOTTLENECK DIAGNOSTICS

We employed three templates exhibiting different levels of perplexity during the generation phase. These templates were divided into three categories (easy, moderate and hard) and labeled with their distinct perplexity scores. The details are presented below.

| **Easy (ppl=13.2)** | **Moderate (ppl=18.2)** | **Hard (ppl=22.6)** |
|---|---|---|
| Consider a **delivery service** that needs to visit 4 cities. The distances (km) are: 

A–B: 184.2, A–C: 71.6, A–D: 94.6 
B–C: 126.8, B–D: 94.5, C–D: 64.0 

Find **the shortest possible route** that visits each city once and returns to the start. | Consider a **routing task** where a planner constructs a tour visiting 4 cities. Pairwise travel distances are **deterministic and symmetric**: 
A–B: 184.2, A–C: 71.6, A–D: 94.6 
B–C: 126.8, B–D: 94.5, C–D: 64.0 

Identify a **minimum-length cyclic route** visiting each city once and returning to the departure. | Consider a **canonical TSP** instance involving 4 nodes in the Euclidean plane. The **symmetric distance matrix** (km) is encoded as: 

A–B: 184.2, A–C: 71.6, A–D: 94.6 
B–C: 126.8, B–D: 94.5, C–D: 64.0 

Compute a **shortest Hamiltonian tour**, minimizing aggregate travel distance over all legs. |

*Figure 9.* Comparison of Prompt Variation across Three Complexity Tiers. While the underlying TSP structure remains invariant, the linguistic framing is scaled from simple delivery metaphors to formal optimization terminology to evaluate model robustness to description perplexity.

Additionally, Figure 10 details the augmented objective descriptions across various problem types. The corresponding evaluation results, which demonstrate the effectiveness of these specific descriptions, are presented and analyzed in Figure 6.

| **TSP** | **Knapsack** | **Inventory** |
|---|---|---|
| **Augmented objective description:** In addition, due to mandatory detours, the driver must travel an extra fixed distance of 10 kilometers in total, no matter what the route is. | **Augmented objective description:** If the hiker selects at least one item, they receive an additional fixed bonus of bonus value points added to the total value (this bonus is added only once, no matter how many items are selected). | **Augmented objective description:** The total cost consists of four components: the ordering cost (p per unit ordered), the holding cost (h per unit of inventory), the shortage penalty (c per unit unmet demand), **and a fixed operational cost of K, which is incurred regardless of decisions.** |

*Figure 10.* Augmented objective descriptions across different problem types, whose results are shown in Figure 6.

## B. Implementation Details of the OPT-Engine Framework

### B.1. Rephrasing and Validation Prompt Templates

In this section, we present the specific prompt templates employed for the problem rephrasing and validation processes. The **Rephrase Prompt** is designed to generate diverse narrative scenarios while strictly preserving the underlying mathematical structure. Subsequently, the **Rephrase-Verified Prompt** serves as a quality control mechanism to rigorously verify the structural equivalence between the original and the rephrased problems.

---

**Rephrase Prompt**

You are an expert in operations research problem design and NLP data augmentation. Your task is to take the following optimization problem and rewrite it according to the instructions.

Original Problem: {{Original problem}}

Instructions:
- Rewrite the problem in a different real-world scenario or application context, while preserving its mathematical structure, optimization goal, and logical constraints.
- All numerical values, quantities, and parameter relationships must remain exactly the same.
- Use different terminology, phrasing, and narrative style to describe the problem, but ensure that the underlying model and relationships are identical.
- Do not add or remove any mathematical constraints, variables, or objectives.
- The rewritten problem should read naturally and clearly as a self-contained description in the new scenario.
- Do not include any explanations, reasoning, or headers.
- Output only the rewritten problem description, without commentary.

Output: [Start your output below. No headers, no comments.]

---

**Rephrase-Verified Prompt**

You are an expert in operations research and mathematical modeling.
Below are two optimization problem descriptions. Please check if the mathematical structure of the rephrased problem is fully equivalent to the original one — meaning they have the same decision variables, objective function type (min or max), and constraint relationships.

Original Problem: {{Original problem}}

Rephrased Problem: {{Rephrased problem}}

Answer only in the following JSON format:
"equivalent": true/false, "reason": "your short reasoning"

---

### B.2. Classification and Design Templates for Optimization Problems

B.2.1. TRAVELING SALESMAN PROBLEM (TSP)

**Problem Description.** The Traveling Salesman Problem(TSP) is a classical combinatorial optimization problem where a salesman must find the shortest possible routes that visits each city exactly once and return to the original city, given a list of cities and the distances between each pair of cities. In formal terms, the TSP can be represented on a complete weighted graph $G = (V, E)$, where each vertex $v_i \in V$ corresponds to a city, and each edge $(v_i, v_j) \in E$ has an associated distance $d_{ij}$. A binary decision variable $x_{ij}$ is defined such that $x_{ij} = 1$ if the route directly travels from city $i$ to city $j$, and $x_{ij} = 0$ otherwise. The objective is to find a Hamiltonian cycle with the minimum total edge weight: $\min \sum_{(i,j) \in E} d_{ij} x_{ij}$ subject to

constraints ensuring that each city is entered and left exactly once, and subtour elimination constraints to guarantee a single connected tour.

This problem is an NP-hard optimization task whose complexity increases factorially with the number of cities, that is, $n$ cities correspond to $n!$ possible routes. Even modest increases in $n$ lead to a combinatorial explosion in search space, making the problem an effective benchmark for evaluating optimization reasoning under scaling difficulty.

**Problem Design.** We control the problem complexity by varying the number of cities $n$. {*city_lines*} specifies the list of cities along with their corresponding coordinates, while {*distance_text*} describes the pairwise distances between cities. {*example_route*} provides an illustrative example of a possible route, formatted as $A \rightarrow B \rightarrow D \rightarrow C \rightarrow A$.

---

**Traveling Salesman Problem(TSP) Template**

Consider a delivery service that needs to visit {n} cities: {city_lines}. The distances between cities are measured in kilometers:{distance_text}. The goal is to find the shortest possible route that visits each city exactly once and returns to the starting city. This creates a 'tour' that minimizes the total travel distance. For example, with just these n cities, starting from City A, some possible routes are: {example_routes}. The challenge is to determine which of all possible routes has the minimum total distance.

---

B.2.2. BIN PACKING PROBLEM

**Problem Description.** The Bin Packing Problem is a classical NP-hard combinatorial optimization problem that seeks the most efficient way to pack a given set of items into the minimum number of identical bins, each with a fixed capacity. Formally, let there be $n$ items indexed by $i = 1, \ldots, n$, each with weight $w_i$, and bins of identical capacity $C$. Define a binary variable $x_{ij}$ such that $x_{ij} = 1$ if item $i$ is assigned to bin $j$, and $x_{ij} = 0$ otherwise, and a binary variable $y_j$ indicating whether bin $j$ is used. The problem can be formulated as:

$$
\begin{aligned}
\min \quad & \sum_j y_j \\
\text{s.t.} \quad & \sum_j x_{ij} = 1, && \forall i = 1, \ldots, n \quad \text{(Each item in One Bin)} \\
& \sum_i w_i x_{ij} \leq C\, y_j, && \forall j \quad \text{(Bin Capacity Constraint)} \\
& x_{ij} \in \{0,1\}, \quad y_j \in \{0,1\}, \quad \forall i, j.
\end{aligned}
\tag{4}
$$

The objective is to minimize the number of bins used while ensuring that no bin exceeds its capacity and every item is packed exactly once. The Bin Packing Problem is NP-hard, with computational complexity growing exponentially with the number of items $n$. Exact algorithms such as branch-and-bound or integer programming exhibit worst-case complexity of $\mathcal{O}(2^n)$

**Problem Design.** We control the problem complexity by varying the number of items $n$ and the bin capacity $C$. {item_lines} specifies each product's weight, while {bin_capacity} denotes the maximum capacity of each bin. The goal is to pack all items into the fewest possible bins without exceeding their capacity constraints.

---

**Bin packing Problem Template**

A warehouse manager needs to pack different products into identical shipping containers. The available items include: {item_lines}. Each shipping container has a maximum weight capacity of {bin_capacity} kg. The manager's goal is to use the minimum number of containers while ensuring all products are packed. Each product must be assigned to exactly one container, and the total weight in each container cannot exceed its capacity.

---

B.2.3. JOB-SHOP SCHEDULING PROBLEM

**Problem Description.** The Job-Shop Scheduling Problem is a classical combinatorial optimization problem that involves determining the most efficient way to process multiple jobs on multiple machines. Each job consists of a specific sequence of operations, and each operation must be performed on a designated machine for a fixed processing time. Once an operation

starts, it must run continuously until completion, and each machine can process only one operation at a time. The goal is to find a feasible schedule that minimizes the makespan—the total time required to complete all jobs. In formal terms, the problem can be represented by a set of jobs $\mathcal{J}$ and a set of machines $\mathcal{M}$. Each job $j \in \mathcal{J}$ is defined as a sequence of ordered operations $(O_{j1}, O_{j2}, \ldots, O_{jK_j})$, where each operation $O_{jk}$ is associated with a machine $M_{jk} \in \mathcal{M}$ and a processing time $p_{jk}$. The objective is to determine the start time of each operation on its assigned machine such that no two operations overlap on the same machine, and the precedence order of operations within each job is respected.

The jobshop problem is an NP-hard optimization problem whose complexity grows combinatorially with the number of jobs and machines. As both dimensions increase, the number of feasible schedules expands exponentially, making it an effective benchmark for assessing reasoning and optimization performance under increasing problem difficulty.

**Problem Design.** We control the problem complexity by varying the number of jobs $n$ and machines $m$. {job_text} specifies each job's sequence of operations, where each operation is represented as a pair of machine and processing time. The goal is to determine the optimal processing order on all machines that minimizes the makespan, ensuring that each machine processes only one operation at a time and that job precedence constraints are satisfied.

---

**Job-shop Problem Template**

Suppose there are {n_jobs} jobs that need to be processed on {n_machines} machines. Each job consists of a sequence of operations represented as pairs (Machine, Processing time), where each pair specifies the machine on which the operation must run and the amount of time it requires. The order of pairs indicates the required sequence in which the operations must be performed. Job details: {job_text} Each operation must run continuously once it starts and cannot be interrupted, and each machine can only process one operation at a time. The objective is to determine the processing order of all operations on the machines so that the makespan (i.e., the total completion time of all jobs) is minimized.

---

### B.2.4. MINIMUM COST NETFLOW PROBLEM

**Problem Description.** The minimum cost network flow problem in the transportation form seeks the most cost-efficient way to ship goods from a set of warehouse nodes $\mathcal{S}$ (supply nodes) to a set of store nodes $\mathcal{D}$ (demand nodes), while satisfying both supply and demand requirements and respecting capacity limits on each transportation route. Each arc $(i, j)$ from warehouse $i \in \mathcal{S}$ to store $j \in \mathcal{D}$ has an associated unit transportation cost $c_{ij}$, capacity limit $u_{ij}$, and flow variable $x_{ij} \in \mathbb{Z}_+$. Each warehouse $i$ provides a supply amount $s_i$, and each store $j$ requires a demand amount $d_j$, where total supply equals total demand: $\sum_{i \in \mathcal{S}} s_i = \sum_{j \in \mathcal{D}} d_j$. The objective is to minimize the total transportation cost while ensuring all supply and demand constraints are satisfied:

$$
\begin{aligned}
\min_{x_{ij} \in \mathbb{Z}_+} \quad & \sum_{i \in \mathcal{S}} \sum_{j \in \mathcal{D}} c_{ij}\, x_{ij} \\
\text{s.t.} \quad & \sum_{j \in \mathcal{D}} x_{ij} = s_i, \quad \forall i \in \mathcal{S} \quad \text{(Supply Constraints)} \\
& \sum_{i \in \mathcal{S}} x_{ij} = d_j, \quad \forall j \in \mathcal{D} \quad \text{(Demand Constraints)} \\
& 0 \leq x_{ij} \leq u_{ij}, \quad \forall (i, j) \in \mathcal{S} \times \mathcal{D} \quad \text{(Capacity Constraints)}.
\end{aligned}
\tag{5}
$$

The computational complexity grows rapidly with the number of nodes and arcs, since the total number of decision variables scales with $|\mathcal{S}| \times |\mathcal{D}|$. As the network expands, the solution space increases combinatorially, making the optimization problem more challenging for larger instances.

**Problem Design.** We control the complexity of the problem by varying the total number of nodes $n$, which determines the number of warehouses and stores in the network. {warehouse_lines} specifies the supply capacity of each warehouse, {store_lines} specifies the demand of each store, and {arc_lines.strip()} describes the transportation routes between them, including each route's capacity limit and per-unit shipping cost.

---

> **Netflow Problem Template**
>
> A logistics company needs to ship goods from {n} warehouses to {n} retail stores: Each warehouse has a supply capacity: {warehouse_lines}. Each retail store has a fixed demand: {store_lines}. The transportation routes between each warehouse and store have specific capacity limits and shipping costs (cost per unit): {arc_lines.strip()}. The company wants to determine how many units of goods to ship from each warehouse to each store in order to minimize the total shipping cost, while satisfying all store demands, not exceeding any warehouse's supply, and respecting the capacity limits of each transportation route.

### B.2.5. KNAPSACK PROBLEM

**Problem Description.** The Knapsack Problem is a classical combinatorial optimization problem where, given a set of items each with a weight and a value, the goal is to determine which items to include in a collection so that the total weight does not exceed a given capacity limit, while maximizing the total value obtained. In formal terms, let each item $i \in \{1, 2, \ldots, n\}$ have a value $v_i$ and a weight $w_i$, and let $W$ denote the maximum weight capacity of the knapsack. Define a binary decision variable $x_i \in \{0, 1\}$, where $x_i = 1$ if item $i$ is included in the knapsack, and $x_i = 0$ otherwise. The problem can then be formulated as:

$$\max \sum_{i=1}^{n} v_i x_i \quad \text{s.t.} \quad \sum_{i=1}^{n} w_i x_i \leq W, \quad x_i \in \{0, 1\}, \ i = 1, \ldots, n. \tag{6}$$

The computational complexity of the Knapsack Problem grows exponentially with the number of items $n$ when solved by exhaustive enumeration, and pseudo-polynomially with the product of the number of items and the capacity $W$ when solved using dynamic programming ($O(nW)$). Consequently, increasing either $n$ or $W$ significantly amplifies the computational burden.

**Problem Design.** We control the problem complexity by varying the number of items $n$, and define the knapsack capacity as a fixed ratio of the total weight of all items. {*item_list*} specifies the list of items, each associated with a weight and a value.

> **Knapsack Problem Template**
>
> A hiker is preparing for an outdoor hiking trip. They need to select the most valuable combination of equipment and supplies from many available options within the limited backpack capacity. The items include: {items_list}. Assuming the backpack has a maximum weight capacity of {capacity} kg, the hiker's goal is to select the combination of items with the highest total value while not exceeding the weight limit. Each item must be either taken in its entirety or left behind.

### B.2.6. INVENTORY PROBLEM

**Problem Description.** The Inventory Problem is an optimization-based decision-making problem based on coordinating procurement, storage, and shortage management over multiple periods. The problem considers a planning horizon of $T$ discrete periods. In each period $t$, the decision-maker determines the quantity of orders, subject to the daily quantity limit $Q_{\min}$ and $Q_{\max}$, given the unit purchase cost $p$, the holding cost $h$, and the shortage cost $c$. The initial inventory is $I_0$, and the warehouse capacity is denoted by $C$. The goal is to satisfy the time-varying demand $D_t$ in each period $t$, accounting for a delivery lead time $l$, while minimizing the total costs, including purchasing, holding, and shortage costs across the entire horizon.

In formal terms, let $o_t$ denote the order quantity placed at the beginning of period $t$, $a_t$ represent the quantity received at the beginning of period $I_t$ denote the last of inventory amount at the end of period $t$, and $s_t$ represent the shortage during period

$t$. The problem can be formulated as:

$$
\begin{aligned}
\min \quad & \sum_t (po_t + hI_t + cs_t) \\
\text{s.t.} \quad & Q_{\min} \leq o_t \leq Q_{\max}, && \forall t \in T \quad \text{(Order Quantity Constraint)} \\
& I_t = I_{t-1} + a_t + s_t - D_t, && \forall t \in T \quad \text{(Production-Demand Balancing Constraint)} \\
& I_{t-1} + a_t \leq C, && \forall t \in T \quad \text{(Warehouse Capacity Constraint)} \\
& a_t = \left\{ \begin{array}{ll} o_{t-l} & t > l \\ 0 & t \leq l \end{array} \right., && \forall t \in T \quad \text{(Definition of the Receipt Quantity)} \\
& I_0 = I_0, && \text{(Boundary Constraint)}.
\end{aligned}
\tag{7}
$$

The computational complexity of the Inventory Problem is primarily driven by the length of the planning horizon $T$ and the size of the discrete state and action spaces.

In dynamic programming sight, under the warehouse's capacity $C$, there are $O(C)$ states in each period, and the DP recursion updates every state by evaluating all feasible order quantities. The resulting time complexity is therefore $O(T \cdot C \cdot |\mathcal{A}|)$, (where $|\mathcal{A}|$ is the number of admissible order quantities per period), which is pseudo-polynomial in the planning horizon and the sizes of the state and action spaces. Consequently, holding the capacity and the admissible order range, increasing the horizon length $T$ significantly amplifies the computational burden of solving the Inventory Problem.

**Problem Design.** We control the problem complexity by increasing the period $T$, and define the other variables as fixed. {*demand_list*} specifies the list of time-varying demand.

> **Inventory Problem Template**
>
> A factory must develop an ordering and inventory plan for a key material over a planning horizon of T days. The initial inventory at the beginning of the planning period is {I0} units. In each period t = 1, ..., {T}, the supplier allows the factory to place an order whose quantity must lie between {Qmin} and {Qmax} units. However, each order placed will take {lead} day(s) to arrive before it can be used to satisfy demand or replenish inventory. The demand for the material in each period is given as follows: {demand_lines} Shortages are permitted, but any unmet demand will not be back-ordered. Throughout the planning horizon, material quantities are allowed to be fractional, and the total amount of on-hand inventory at any time must not exceed the warehouse capacity of {C} units. The total cost over the planning horizon consists of three components: the ordering cost, which equals {p} per unit ordered; the holding cost, which equals {h} per unit of inventory carried from one period to the next; and the shortage penalty, which equals {c} per unit of unmet demand. Please determine the optimal order quantity for each period and track the resulting inventory and shortage levels so as to minimize the total cost.

### B.2.7. PORTFOLIO ALLOCATION PROBLEM

**Problem Description.** The Portfolio Allocation Problem is a classical Linear Programming problem that evaluates an investor's ability to balance risk, return, and liquidity under multiple investment constraints. The problem considers a set of $I$ asset categories, each characterized by an expected return $r_i$ and an associated risk level $v_i$. The decision-maker determines the investment proportion $x_{ij} \geq 0$ allocated to each asset $i$, which must lie within a specified bound $[l_i, u_i]$. A subset of assets $\mathcal{L} \subseteq I$ represents liquid assets that contribute to the portfolio's liquidity requirement. The goal is to maximize the

portfolio's total expected return while satisfying several constraints. The problem can be formulated as:

$$
\begin{aligned}
\max \quad & \sum_{i \in I} r_i x_i \\
\text{s.t.} \quad & \sum_{i \in I} x_i = 1, && \text{(Budget Constraint)} \\
& \sum_{i \in I} v_i x_i \leq V_{\max}, && \text{(Risk Control Constraint)} \\
& \sum_{i \in I} r_i x_i \geq R_{\min}, && \text{(Minimum Return Constraint)} \\
& \sum_{i \in \mathcal{L}} x_i \geq L_{\min}, && \text{(Liquidity Constraint)} \\
& l_i \leq x_i \leq u_i, && \text{(Investment Proportion Constraint)}.
\end{aligned}
\tag{8}
$$

**Problem Design.** We control the complexity of the problem by increasing the number of categories of assets $I$. $\{asset\_lines\}$ includes the expected return $r_i$, risk level $v_i$ and proportion bounds $l_i, u_i$ of each asset $i$. $\{L\_assets\}$ is a set of all liquid assets. $\{L_{\min}\}$ is the minimum level of liquidity, $\{R_{\min}\}$ is the required minimum return, $\{V_{\max}\}$ is the supreme risk level.

---

**Portfolio Allocation Problem Template**

An investor wishes to allocate capital among {I} asset classes with the goal of maximizing the total expected return of the portfolio. The characteristics of each asset are summarized as follows: {asset_lines}. Each asset must receive a proportion of the total investment that satisfies its individual lower and upper bounds, and the total of all investment proportions must sum to one. To ensure sufficient liquidity, the investor requires that the group of liquid assets, represented by the subset L ={L_assets}, collectively receive at least {L_min} of the total capital. At the same time, the overall portfolio risk, measured by a specified risk index, must not exceed {V_max}, and the total expected return of the portfolio must be no less than {R_min}. Please determine the optimal portfolio weights that maximize total expected return subject to all constraints.

---

### B.2.8. PRODUCTION PROBLEM

**Problem Description.** The Production Problem is a classical linear programming formulation that seeks to maximize total profit subject to limited resource constraints. Consider a simplified setting with $n$ types of products to be manufactured, each requiring $m$ distinct production processes. For convenience, we assume that the types of products and the number of processes are equal(that is $m = n$). The profit per kilogram of product $i$ is denoted by $p_i$, and the time required for product $i$ in process $j$ is represented by $t_{ij}$. Each process $j$ has a maximum available processing time of $T_j$. The objective is to determine the optimal production quantities that maximize the overall profit. Accordingly, the problem can be formulated as follows:

$$
\begin{aligned}
\max \quad & \sum_i p_i x_i \\
\text{s.t.} \quad & \sum_i t_{ij} x_i \leq T_j, \quad \forall j \quad \text{(Time Constraint)} \\
& x_i \geq 0, \qquad\qquad \forall i \quad \text{(Non-Negative Constraint)}.
\end{aligned}
\tag{9}
$$

**Problem Design.** We control the problem complexity by varying the number of product types and processing operations. The set $\{I\}$ represents the collections of products and production processes, respectively, whose sizes are equal. The term $\{unit\_label\}$ denotes the unit of measurement for each product. The specification $\{profit\_lines\}$ provides the profit values associated with each product. Both $\{time\_lines\}$ and $\{op\_cap\_lines\}$ describe the processing time required for each product across different operations, while $\{op\_range\}$ indicates the maximum available time capacity for each operation.

---

**Production Problem Template**

A factory intends to produce {I} types of products, each of which requires {I} processing operations to complete. The profit earned per {unit_label} of each product is given as follows: {profit_lines}. Processing time required for each operation is as follows: {time_lines}, {op_cap_lines}. On the premise of guaranteeing for each operation {op_range}, total processing time must not exceed its available time. Please schedule the production plan to maximize total profit.

---

### B.2.9. TRANSPORTATION PROBLEM

**Problem Description.** The transportation problem is a classical linear optimization problem that focuses on minimizing total shipping cost while satisfying supply and demand constraints across multiple locations. It serves as a fundamental model in operations research and logistics optimization, widely applied in production planning, distribution management, and resource allocation.

The problem involves two disjoint sets of nodes: a set of production sites $A = \{1, 2, \cdots, n\}$ and a set of sales destinations $B = \{1, 2, \cdots, m\}$. Each production site $A_i$ has a limited supply capacity $e_i$, while each sales destination $B_j$ has a fixed demand requirement $d_j$. The cost of transporting one unit of goods from production site $A_i$ to destination $B_j$ is denoted by $c_{ij}$. The decision variable $x_{ij}$ represents the amount of goods shipped from $A_i$ to $B_j$.

The objective is to determine an optimal shipping plan that minimizes the total transportation cost. The problem can be formulated as

$$
\begin{aligned}
\min \quad & \sum_i \sum_j c_{ij} x_{ij} \\
\text{s.t.} \quad & \sum_j x_{ij} \leq e_i, \quad \forall i \in A \quad \text{(Supply Constraint)} \\
& \sum_i x_{ij} = d_j, \quad \forall j \in B \quad \text{(Demand Constraint)} \\
& x_{ij} \geq 0, \qquad \forall i, j \in A, B \quad \text{(Non-Negative Constraint)}.
\end{aligned}
\tag{10}
$$

Each production site in set $A$ is associated with a fixed supply capacity, while each sales destination in set $B$ has a specified demand that must be satisfied exactly. For convenience, we control both sets' size in $A$. The unit shipping cost between each production site and destination is given by a cost matrix $C = [c_{ij}]$. By adjusting $n$, we can scale the dimensionality of the decision space and the number of flow constraints.

**Problem Design.** We control the problem complexity by varying the number of production sites (which equals the number of sales destinations) $n = |A|(|B|)$. The specification {*supply_lines*} presents the available supply quantities for each production site in set $A$, while {*demand_lines*} provides the required demand levels for each destination in set $B$. The term {*cost_lines*} denotes the unit transportation cost matrix between all production–destination pairs. In particular, the cost information is organized and displayed in a tabular form for clarity and ease of reference.

---

**Transportation Problem Template**

Consider a transportation problem that aims to minimize the total shipping cost from production sites A to sales destinations B. The available supply at each production site (set A) is given as follows: {supply_lines}. The demand that must be met at each sales destination (set B) is specified below: {demand_lines}. The unit shipping cost from each production site to each destination is as follows: {cost_lines}. Please choose the shipment to minimize the total cost.

---

### B.2.10. POLLUTION CONTROL PROBLEM

**Problem Description.** The pollution control problem is a constrained optimization problem that focuses on minimizing the total cost of emission control while achieving a predefined pollution reduction target. The problem involves a set of emission sources $T = \{1, 2, \cdots, T\}$, each representing a thermal power plant or industrial facility that emits pollutants such as flue gas. Each source $i$ has an emission rate $w_i$ and a production output $p_i$. A set of available abatement technologies $K = \{1, 2, \cdots, K\}$ is provided, where each technology $j$ is characterized by a removal efficiency $s_j$ and an associated unit

cost $c_{ij}$ when applied to source $i$.

The decision variable $x_{ij}$ represents the quantity of production at emission source $i$ that adopts abatement technology $j$. The goal is to minimize the total abatement cost. The problem can be formulated as,

$$
\begin{aligned}
\min \quad & \sum_{i=1}^{T} \sum_{j=1}^{k} c_{ij} x_{ij} \\
\text{s.t.} \quad & \sum_{j=1}^{k} x_{ij} = p_i && \forall j \quad \text{(Production Constraint)} \\
& \sum_{i=1}^{T} \sum_{j=1}^{k} w_i x_{ij} s_j \geq \mathcal{P} \cdot \sum_{i=1}^{T} w_i p_i && \text{(Pollution Reduction Constraint)} \\
& x_{ij} \geq 0 && \forall i,j \quad \text{(Non-Negative Constraint)}
\end{aligned}
\tag{11}
$$

**Problem Design.** We control the problem complexity by varying the number of emission sources $T$ and available control methods $K$. For convenience, we assume these two numbers are equal($T = K$). {*source_lines*} specifies the characteristics of each emission source, including its emission rate and production output. {*method_lines*} describes the set of available control methods, each associated with a distinct removal efficiency. The cost structure for all source method combinations is summarized in {*cost_lines*}, which is displayed in a tabular form.

---

**Pollution Control Problem Template**

A region seeks to design an air-pollution control plan to reduce total suspended particulate (TSP) emissions from several industrial point sources. Initially, no control measures have been applied. The characteristics of each emission source are as follows: {source_lines}. To mitigate emissions, several control methods are available, each characterized by a specific removal efficiency: {method_lines}. Applying a control method to a source incurs an additional cost per unit of production. The cost structure for all source-method combinations is summarized below: {cost_lines}. Please choose how to apply control methods to each source (sources may adopt multiple methods simultaneously), and note that a source may also remain partially uncontrolled if necessary. The goal is to ensure that the total TSP emissions are reduced by at least proportion P = {P} of E0, while minimizing the total cost.

---

# C. Details of Experiments

## C.1. Experiment Setup

### C.1.1. EVALUATION PROMPT TEMPLATE

To evaluate the solution accuracy of each optimization problem instance, we utilize the following two prompt templates, corresponding to the SIR, PTR and compPTR paradigms, respectively.

---

**Solver-Integrated Reasoning (SIR)**

You are an operation research and Gurobi solver expert. Below is an operations research question:
{{Question}}
Carefully analyze the problem to identify the core elements such as Decision Variables, Objective Function, and Constraints, determine whether the variable is an integer or a continuous variable; Build a mathematical model and corresponding gurobi codes start with:
python
import gurobipy as gp
from gurobipy import GRB
- Make sure the model variable is named 'model'. - Avoid using "<" and ">" in Gurobi constraints; instead, use "<=" or ">=" as appropriate. Think step by step to ensure flawless translation from math to code.

---

**Pure-Text Reasoning (PTR)**

You are a Mathematical Modeling and Optimization Consultant specializing in analytical solutions. Below is an operations research question:
{{Question}}
Your task is to rigorously formulate and solve the following optimization problem.
Follow this structured approach: Begin with understanding the problem → Extract the set and parameters → Identify the variables → Provide the objective function → Analyze the constraints → Develop the mathematical model → Solve it step-by-step, output the corresponding decision variables
Instruct: 1.) All equations and mathematical definitions must be presented clearly. 2.) Provide a clear, step-by-step solution process. Absolutely do not use or reference any external OR solvers or software (e.g., Gurobi, PuLP, Solver). The solution must be purely analytical. Output the final optimal objective function value in markdown with the exact tag: <answer> optimal value here < /answer>

---

**Computational PTR (compPTR)**

You are a Mathematical Modeling and Optimization Consultant specializing in analytical solutions. Below is an operations research question:
{{Question}}
Follow this structured approach: Begin with understanding the problem; Extract the set and parameters ; Identify the variables ; Provide the objective function ; Analyze the constraints ; Develop the mathematical model ; Solve it step-by-step, and output the corresponding decision variables.
Instruct: 1.) You should leverage both natural language reasoning and interactive Python code execution. Your goal is to provide clear, detailed explanations while utilizing Python to perform complex calculations, symbolic manipulations, data analysis, or any other tasks that can aid in problem-solving.
2.) Absolutely do not use or reference any external OR solvers or software (e.g., Gurobi, PuLP, Solver). Your entire response should include Step-by-Step reasoning processes and conclude with a single, self-contained Python code block that integrates all previous logic and performs the final calculation. The code blocks should start with "'python. The script's final output line must be: print("Result:", objective_value) where objective_value is the actual computed result variable.

---

C.1.2. TRAINING AND INFERENCE SETTING

**Training setup.** The training of Qwen3-4B-RL involved a rigorous RL training phase initialized from Qwen3-4B-Instruct (Yang et al., 2025a). Following the Solver-Informed RL approach (Chen et al., 2025), we adapted the Verl framework (Sheng et al., 2025) to incorporate optimization domain-rewards into the advantage estimation. Training was performed on a node with eight 80GB NVIDIA H100 GPUs, requiring 192 GPU hours per stage. The key hyperparameters for Qwen3-4B-RL training are detailed in Table 3:

*Table 3.* Training Parameters

| Type | Parameter | Value |
|---|---|---|
| **Algorithm** | Advantage Estimator | reinforce_plus_plus |
| **Data** | Batch size | 64 |
| | Learning rate | 1e-6 |
| | Max prompt length | 2048 |
| | Max response length | 16384 |
| | Truncation | left |
| **Actor/Rollout** | KL loss type | low_var_kl |
| | KL loss coefficient | 0.005 |
| | Rollout number | 8 |
| | PPO mini batch size | 8 |
| | PPO micro batch Size per GPU | 4 |
| | Clip ratio low | 0.20 |
| | Clip ratio high | 0.28 |

**Inference Setup.** We employ the top-P (nucleus) decoding strategy (Holtzman et al., 2020) for the training and inference phases. The exact sampling hyperparameters used to generate our main results are specified in Table 4:

*Table 4.* Sampling parameters used for text generation.

| Parameter | Value |
|---|---|
| n | 1 |
| Temperature | 0.5 |
| Top-p | 0.95 |
| Max Response Length | 16128 |
| Repetition penalty | 1.02 |

## C.2. Experimental Details of SIR and PTR Comparison

### C.2.1. EXPERIMENTAL DETAILS FOR TEN PROBLEMS CLASSES

We evaluated ten problem classes in OPT-ENGINE using DeepSeek V3.2, GPT 5.1, Qwen3-4B Instruct and its fine-tuned variant Qwen3-4B-RL, applying both the SIR and PTR methods. As the results for the Traveling Salesman Problem (TSP), the knapsack Problem, and the inventory problem are presented in the main text, the figures for the remaining seven problem classes are detailed below. For SIR, we interfaced with the Gurobi solver using a 100-second execution limit; no timeouts occurred as all instances fell within this computational threshold.

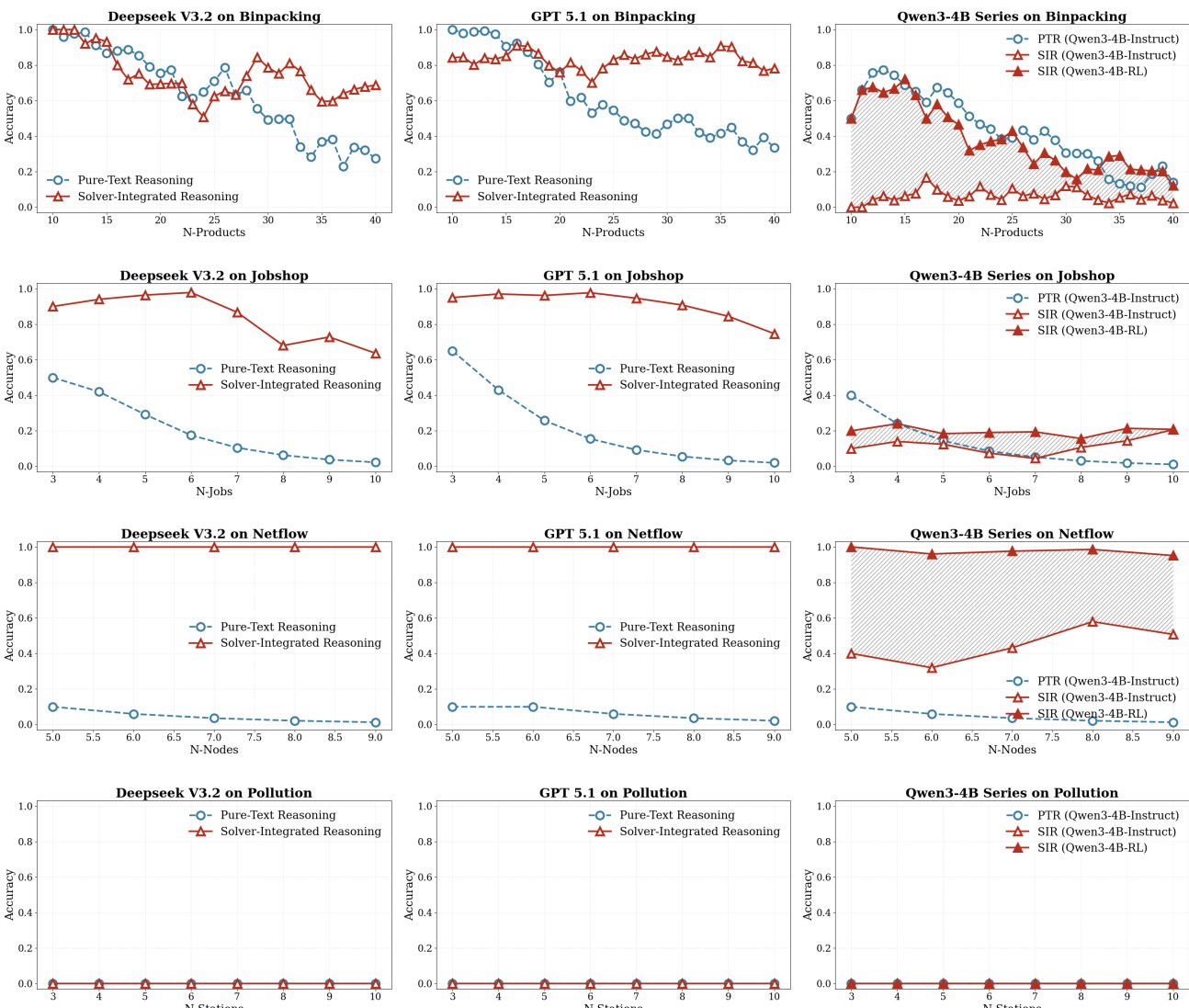

*Figure 11.* Performance comparison of Solver-Integrated Reasoning (SIR) and Pure-Text Reasoning (PTR) across optimization tasks (Part 1). The first column shows DeepSeek V3.2 results; the second and third columns present GPT-5.1 and Qwen3-4B Models under identical settings. Problems: bin packing, job-shop scheduling, network flow, pollution control.

### C.2.2. EXPERIMENTS DETAILS ON PUBLIC DATASETS

We also evaluated the SIR and PTR methods on three public available datasets: Mamo Complex, IndustryOR, and OptMATH. The results are presented below.

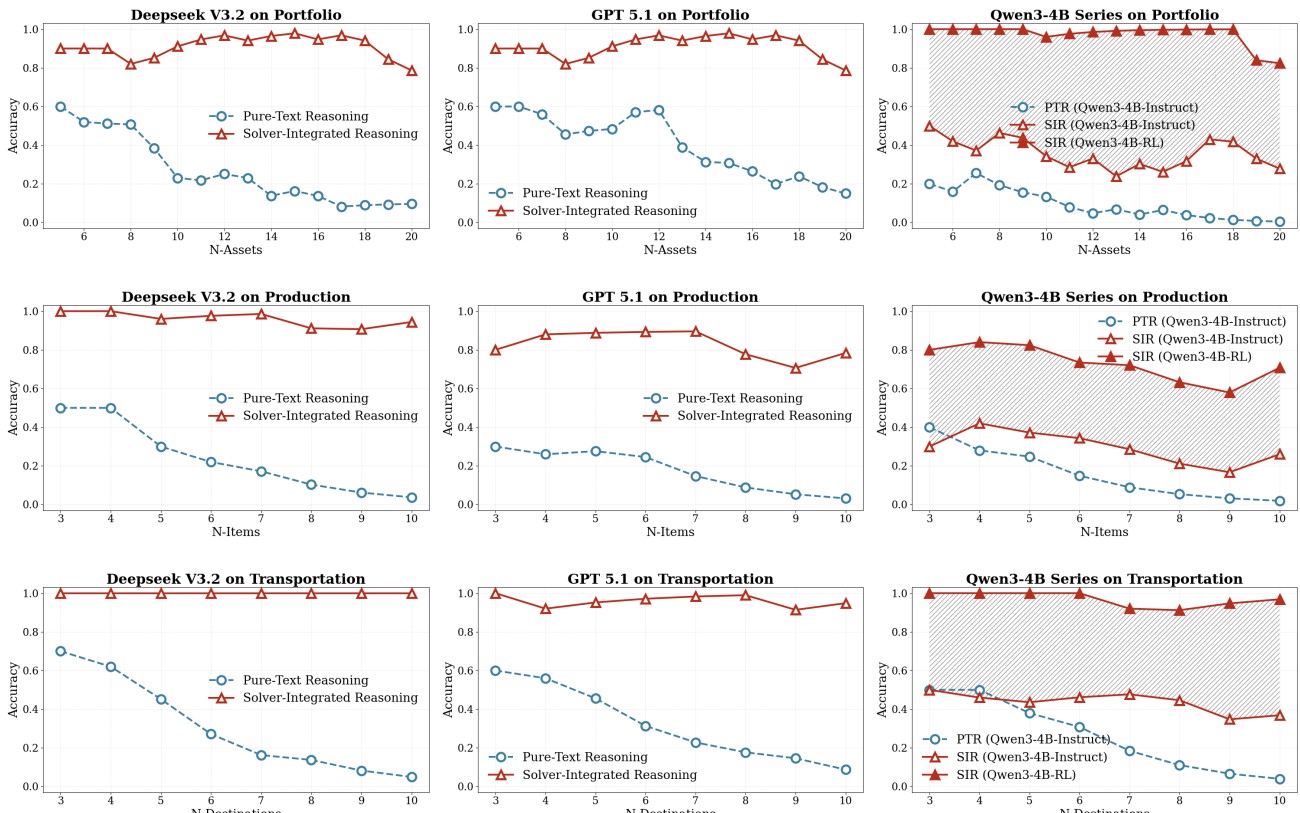

*Figure 12.* Performance comparison of SIR and PTR across optimization tasks (Part 2). Model arrangement identical to Part 1. Problems: portfolio allocation, production planning, transportation.

| Base Model | MamoComplex | | IndustryOR | | OptMATH | |
|---|---|---|---|---|---|---|
| | PTR | SIR | PTR | SIR | PTR | SIR |
| DeepSeek-V3.2 | 38.9% | 56.2% | 49.0% | 61.0% | 21.1% | 36.1% |
| GPT-5.1 | 54.7% | 61.1% | 54.0% | 65.0% | 24.1% | 35.5% |
| Qwen3-4B-Instruct | 38.9% | 17.7% | 49.0% | 18.0% | 10.2% | 10.2% |
| Qwen3-4B-RL | 39.4% | 49.8% | 55.0% | 51.0% | 25.3% | 30.7% |
| Δ | +0.5% | +32.1% | +6.0% | +33.0% | +15.1% | +20.5% |

*Table 5.* Comparison between SIR and PTR across established benchmarks. The top two rows feature frontier API-based models (DeepSeek-V3.2 and GPT-5.1), while the bottom two rows display the compact Qwen3-4B-Instruct and its fine-tuned variant, Qwen3-4B-RL.

C.2.3. CASE STUDY: PTR OUTPUT FOR TSP

We examine DeepSeek-V3.2's Pure-Text Reasoning (PTR) traces on the Traveling Salesman Problem (TSP). By contrasting two instance scales ($N = 4$ and $N = 8$), we demonstrate how increased dimensionality shifts the model's reasoning strategy: while it performs an explicit enumeration of Hamiltonian cycles for small-scale instances, it reverts to a greedy nearest-neighbor heuristic as the search space expands.

| TSP (N-cities=4) | TSP (N-cities=8) |
|---|---|
| With $n = 4$, possible **Hamiltonian cycles**: $\frac{(n-1)!}{2} = \frac{3!}{2} = 3$ distinct tours (undirected). Let's list them and compute total distances. | We can attempt a **Nearest Neighbor Heuristic** for an initial solution, then refine it manually. |
| **Tour 1:** $1-2-3-4-1$  $(1-2) : 74.3$  $(2-3) : 26.0$  $(3-4) : 116.1$  $(4-1) : 168.4$  **Sum:** 384.8 km | **Step 1:** List distances. A–B: 147.1, A–C: 163.7, … |
| **Tour 2:** … | **Step 2:** Start at A. Nearest neighbor: D (44.3). From D, the closest unvisited is E (82.1). Tour: A–D–E … |
| **Tour 3:** … | **Step 3:** Check improvements. Ending with F–A (165.4) is suboptimal. |
| **Minimum:** 384.8 km. | **Best found:** 502.7 km. |

*Figure 13.* Comparative Analysis of DeepSeek-V3.2 Performance in Pure-Text Reasoning for TSP: $N = 4$ vs. $N = 8$

# D. Additional Experiments on Primary Bottleneck Identification

## D.1. Experimental Results on GPT 5.1

We conducted the three experiments described in Section 5 using DeepSeek V3.2 and GPT-5.1. While the results obtained with DeepSeek V3.2 are presented in the main text, the corresponding results for GPT-5.1 are reported below.

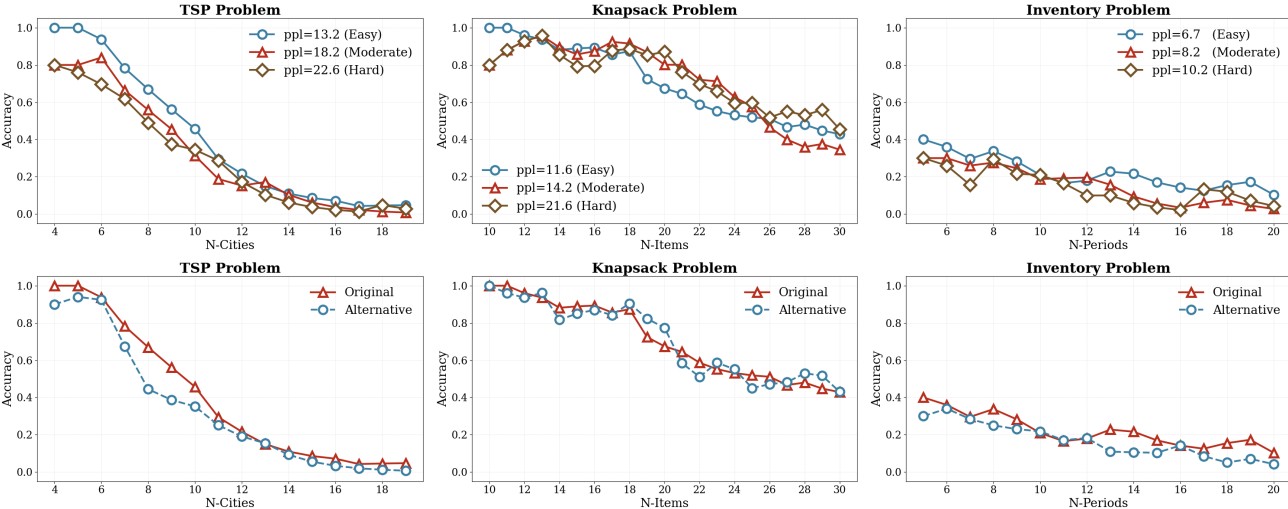

*Figure 14.* **Performance scaling on the GPT 5.1 Model**. **Top Row**: Accuracy under varying linguistic complexity (Easy, Moderate, and Hard tiers, measured by PPL); **Bottom Row**: Accuracy comparison between the original and perturbed objective functions.

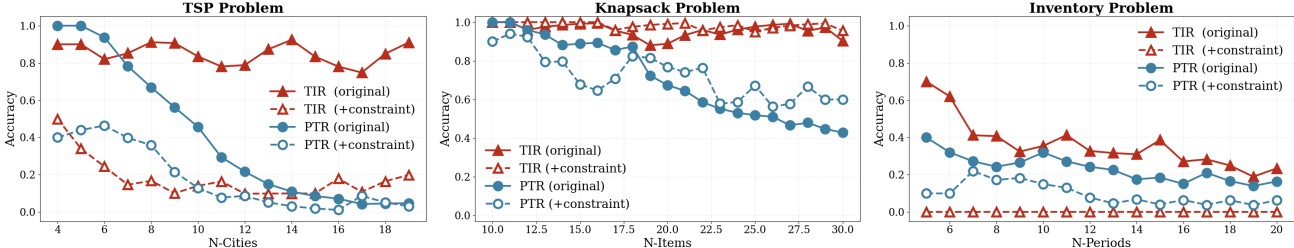

*Figure 15.* **Performance scaling under baseline vs. augmented constraints on the GPT 5.1 Model.** PTR is shown in blue and SIR in red. Augmented constraints used in evaluation are as follows: **TSP**: Exactly one of the following two roads must be included in the tour: the road between city 1 and city 2, the road between city 2 and city 3. ($x_{01} = 0$, $x_{12} + x_{23} = 1$); **Knapsack**: Exactly one of item 1 and item 2 must be selected. If item 3 is selected, the effective backpack capacity is reduced by 2 kg. ($x_1 + x_2 = 1$, $\sum_{i=1}^{n} w_i x_i \leq C - 2x_3$); **Inventory**: On day $t$, the on-hand inventory must be at least $I_{\min}$ units. ($I_t \geq I_{\min}$). The introduction of augmented constraints leads to a consistent accuracy drop across problem classes.

