# OpenReview forum: "OPT-Engine: Benchmarking the Limits of LLMs in Optimization Modeling via Complexity Scaling"
_ICML.cc/2026/Conference — ICML 2026 regular_

### Official Review · Reviewer_eFPH · 2026-02-14

**Soundness:** 4
**Presentation:** 4
**Significance:** 3
**Originality:** 4
**Overall Recommendation:** 4
**Confidence:** 4

**Summary:**

This paper introduces OPT-Engine, an extensible benchmark framework designed to evaluate the capabilities and scalability of Large Language Models (LLMs) in optimization modeling. Unlike existing benchmarks that often rely on static, textbook-style problems, OPT-Engine systematically generates problem instances across ten canonical Operations Research (OR) classes (spanning both Linear Programming and Mixed-Integer Programming) with controllable complexity. The framework employs a four-stage pipeline—numeric generation, problem construction, augmentation, and validation—to create diverse, verifiable instances backed by ground-truth solutions from solvers like Gurobi.

The authors utilize this framework to conduct a rigorous comparative analysis between two reasoning paradigms: Tool-Integrated Reasoning (TIR), where LLMs generate code for external solvers, and Pure-Text Reasoning (PTR), where LLMs attempt end-to-end deductive reasoning. The results demonstrate a sharp performance degradation for PTR as problem complexity scales, whereas TIR remains significantly more robust. Furthermore, through diagnostic experiments, the authors identify that constraint formulation—specifically the ability to handle non-canonical, augmented constraints—is the primary bottleneck for current models, rather than linguistic complexity or objective perturbations. The paper concludes that integrating external tools is currently essential for industrial-scale optimization and provides a roadmap for addressing the semantic sensitivity of LLMs to constraint variations.

**Compliance With Llm Reviewing Policy:**

Affirmed.

**Final Justification:**

I think the authors have fully addressed my comment and I will keep my current positive assessment to this paper.

**Key Questions For Authors:**

- Does test-time compute (e.g., majority voting, self-correction, or best_of_n) improve performance, particularly for the PTR baseline? The current results appear to rely on single-pass generation?

- For problems with large parameters, is the primary failure mode data hallucination (forgetting numbers) or a fundamental inability to model the logic? (Section 5 implies it is constraint formulation, but does this hold for simply large but canonical problems?)

- Did you perform any manual human validation on a subset of the generated data? While the automated pipeline checks for feasibility, manual review is often necessary to catch ambiguity or multiple valid interpretations in the natural language descriptions.

- You attribute the PTR failure to a lack of "exact algorithmic computations". Is there any evidence that allowing the model to "output solver code" or use any intermediate domain-specific languages would bridge this gap?

(I would consider raising the score if the authors can address my questions)

**Limitations:**

Another limitation or concern is that the comparison between TIR and PTR might be unbalanced. It is well-established that LLMs, as the next-token predictors, are fundamentally unsuited for the heavy arithmetic and algorithmic state-tracking required for large-scale optimization (especially since they're NP-Hard problems). Expecting a model to solve a bin-packing problmem via chain-of-thoughts will definitely fail as complexity scales.

Maybe a fairer evaluation of the model's intrinsic tool-calling capabilities would be an agentic system equipped with basic tools (e.g., a Python calculator for arithmetic or a scratchpad for state tracking) or a multi-turn refinement workflow. Comparing single-pass CoT directly against a Gurobi-backed system reveals less about "reasoning" limits and more about the known limitations of arithmetic in transformers.

**Strengths And Weaknesses:**

**Strengths**

- The benchmark’s design, which systematically scales problem difficulty via quantifiable parameters rather than relying on static datasets, is a novel and valuable contribution.

- The comparative evaluation of Tool-Integrated Reasoning (TIR) and Pure-Text Reasoning (PTR) effectively mirrors recent advancements in LLM capabilities, validating the necessity of tool-calling for complex logic.

- The paper is very well-presented, with clear writing and effective visualizations of scaling trends (such as Fig. 3) that makes the methodology and results easy to follow

**Weakness**

- While the benchmark scales problem size ($N$), it does not adequately control for formulation complexity. The number of distinct constraint types is often a more critical dimension of difficulty than variable count. Although the authors tried constraint augmentation, these binary constraints do not fully capture the complex, logical constraints found in real-world scenarios.

- The scale and variety of problems do not reach industrial levels. For instance, a network flow problem with thousands of cities requires fundamentally different formulation strategies compared to the 20-city instances presented here. Therefore, this benchmark may not truly predict performance on real-world large-scale tasks.

- In Section 4.3, the authors penalize the model for switching from enumeration to heuristics on larger TSP instances. However, enumeration is an inefficient strategy that should be abandoned as $N$ scales. Expecting an LLM to solve NP-hard problems exactly in pure text seems unrealistic. The evaluation should consider the quality of approximate solutions rather than treating non-exact heuristic solutions strictly as failures.

---

> ### Author Rebuttal · Authors · 2026-03-30
>
> **Q‑1: Evaluation of test‑time approaches and evaluation criteria**
>
> In the manufacture, we use **avg@10** (mean success rate over 10 instances per class/complexity level).
>
> To test if test-time compute improves performance, we conducted additional experiments with self-consistency and reflection under the same **avg@10** protocol (using Deepseek V3.2 model)
>
> |Model|Method|TSP|Bin|Inv|MeanΔ|
> |-|-|-|-|-|-|
> |PTR|Zero-shot|14.71%|51.94%|15.00%|—|
> |PTR|SC(k=8)|27.65%|70.97%|25.62%|+14.20%|
> |PTR|Refl(1)|32.94%|66.45%|26.88%|+14.87%|
> |TIR|Zero-shot|85.09%|72.90%|36.25%|+37.80%|
>
> Results suggest that while test‑time computing methods consistently boost the performance across all classes, the zero-shot TIR still maintains a clear margin over all PTR approaches.
>
> ---
>
> **Q-2, Q-4, W-3 & Limitations: PTR Failure Modes & Exploration of Computation‑Enhanced PTR**
>
> We are sincerely grateful to the reviewer for this thoughtful suggestion. It significantly strengthened the paper by prompting both a more fine-grained diagnosis of PTR/TIR failure modes and the addition of a **"computation‑enhanced" PTR** baseline.
>
> We addressed these questions by:
>    1. Extending current evaluation metrics to include **Feasibility rate/Accuracy rate (F/A)** and **optimality gap**, which provide a more fine-grained view of failure modes.
>    2. Implementing **Agentic-PTR**, where the LLM is prompted to combine natural-language reasoning with Python-based calculation tools to assist intermediate computations and verify steps.
>
> | Problem | **Paradigm** | Easy |  | Medium |  | Hard |  |
> | - | - | - | - | - | - | - | - |
> |  |  | F/A (%/%) | Gap (%) | F/A (%/%) | Gap (%) | F/A (%/%) | Gap (%) |
> | TSP | TIR | 88.3/88.3 | 0.00 | 84.3/84.3 | 0.01 | 82.0/82.0 | 0.00 |
> |  | PTR | 97.1/79.1 | 0.91 | 92.0/11.0 | 8.47 | 80.0/4.0 | 13.12 |
> |  | **Agentic PTR** | 85.0/85.0 | 0.01 | 72.0/70.0 | 0.04 | 52.0/44.0 | 0.43 |
> | Knapsack | TIR | 96.3/96.3 | 0.00 | 91.7/91.7 | 0.00 | 95.7/95.7 | 0.00 |
> |  | PTR | 86.3/82.5 | 0.19 | 63.3/63.3 | 0.00 | 57.1/47.1 | 0.42 |
> |  | **Agentic PTR** | 88.8/88.8 | 0.00 | 83.3/83.3 | 0.00 | 78.6/78.6 | 0.00 |
> | Netflow | TIR | 100.0/100.0 | 0.00 | 100.0/100.0 | 0.00 | 100.0/100.0 | 0.00 |
> |  | PTR | 70.0/10.0 | 0.12 | 60.0/0.0 | 0.13 | 55.0/0.0 | 0.71 |
> |  | **Agentic PTR** | 90.0/90.0 | 0.00 | 85.0/85.0 | 0.00 | 75.0/75.0 | 0.00 |
> | Production | TIR | 96.7/96.7 | 0.00 | 100.0/100.0 | 0.00 | 90.0/90.0 | 0.00 |
> |  | PTR | 33.3/33.3 | 0.01 | 20.0/10.0 | 0.63 | 20.0/0.0 | 2.68 |
> |  | **Agentic PTR** | 66.7/66.7 | 0.00 | 70.0/70.0 | 0.00 | 66.7/66.7 | 0.00 |
>
> These results clarify the distinct failure modes of three paradigms.
>
> - **TIR:** accuracy is tied to feasibility: once the model captures correct logical structures, the solver reliably handles the optimization. So failures mainly stem from mis-formulating constraints or objectives.
>
> - **PTR:** two limitations emerge: **reasoning collapse**, where feasibility drops sharply as difficulty increases because the model struggles to optimize in-context while preserving global mathematical structure, and **calculation limitations** create a large feasibility–accuracy gap, so PTR can produce plausible answers yet still miss the exact optimum (e.g., medium-scale TSP: 92.0% feasible vs. 11.0% accurate).
>
> - **Agentic-PTR:** improves substantially over PTR in both feasibility and accuracy, suggesting that external computational support helps reduce in-context reasoning failures, not just arithmetic errors. But it still underperforms **TIR** because **reasoning collapse** remains reflected in feasibility.
>
> **References**
> [1] "Tool-Star: Empowering LLM-Brained Multi-Tool Reasoner via Reinforcement Learning," 2025.
>
> ---
> **Q‑3: Manual Human Validation**
>
> Yes. The pipeline starts from expert-designed instance-construction code and prompt templates for each problem class, which require substantial time and effort to ensure that original problems intact. In addition to automated validation, we performed **manual validation** primarily on the augmentation stage, with particular attention to semantic clarity and numeric consistency between the generated instance and its natural-language description.
>
> ---
> **Weakness 1 & 2**
>
> We agree **formulation complexity** is not fully captured by variable count, and our current constraint augmentation is a controlled first step rather than a full representation of real-world logical structure. However, even under this relatively simple setting, we already observe a clear performance drop, highlighting current LLMs' sensitivity to non-canonical constraints.
>
> We also agree the current benchmark does not yet reach **industrial scale**. As noted in our response to **Reviewer 1o4E, Weakness 2 & 4**, we additionally extended TSP to 45 cities. Although still not industrial scale, the same qualitative trend persists, with PTR degrading much faster than TIR.
>
> Due to character limitations, we are happy to address additional questions during the discussion period.

---

> > ### Author Rebuttal · Reviewer_eFPH · 2026-04-03
> >
> > Thank you to the authors for their response. After reading the authors' rebuttal and the comments from the other reviewers, I believe it is appropriate to maintain my current score.

---

> > > ### Author Response · Authors · 2026-04-04
> > >
> > > We sincerely appreciate the reviewer's positive and constructive feedback. Your suggestion to examine exact algorithmic computations in PTR approaches prompted a valuable extension of our empirical work.
> > >
> > > Our results confirm that although integrating computational tools and iterative execution enhances local accuracy, **the enhanced PTR reasoning still struggles to satisfy the global optimization constraints** inherent in using LLMs for optimization tasks. This finding offers valuable insights for the LLM-for-optimization modeling community.
> > >
> > > We will integrate these insights into the revised manuscript and believe they will greatly enhance the work. We appreciate your impactful guidance.

---

### Official Review · Reviewer_1o4E · 2026-03-12

**Soundness:** 3
**Presentation:** 3
**Significance:** 2
**Originality:** 3
**Overall Recommendation:** 4
**Confidence:** 4

**Summary:**

The paper introduces OPT-ENGINE, a new benchmark framework designed to evaluate the capabilities of LLMs in optimization modeling by systematically scaling problem complexity. It first compares two primary LLM-based approaches for solving optimization problems: Pure-Text Reasoning (PTR) and Tool-Integrated Reasoning (TIR). Then, primary bottlenecks of current LLMs in optimization are systematically diagnosed through linguistic complexity, objective perturbations, and constraint augmentation. Overall, this paper provides valuable insights for developing next-generation LLMs for optimization modeling.

**Compliance With Llm Reviewing Policy:**

Affirmed.

**Final Justification:**

The author responses address most of my concerns. I decide to raise my score to 4.

**Key Questions For Authors:**

* For `Paradigms in LLM-driven Optimization` in Section 2, there is a line of work that leverages LLMs to evolve code for solving optimization problems within (meta-)heuristic frameworks. How does this line of work fit into your categorization? Why not further benchmark this line of work?
* Could the authors provide examples of objective perturbations and constraint augmentation in the Appendix (similar to Fig. 8)?

**Limitations:**

Yes, limitations are discussed in Section 6.

**Strengths And Weaknesses:**

**Strengths:**
* This paper is well-written and easy to follow.
* Benchmarking LLMs in optimization modeling via *complexity scaling* is a new and timely contribution.
* OPT-Engine covers 10 OR problem classes, ranging from LP to MIP.
* The diagnosis of the primary bottleneck of current LLMs in optimization (in Section 5) is insightful.

**Weaknesses:**
* The font type does not appear to conform to the ICML submission template. I will leave it to the AC to determine whether this violates the requirements of the ICML CFP.
* I would like to see a more comprehensive comparison and discussion clarifying the differences between OPT-Engine and the previous benchmarks listed in Section 2 (“Benchmarks for Optimization Modelling”). For example, the authors could include a table to systematically compare key aspects.
* Although introducing complexity scaling is good, the way of scaling complexity is problem-specific and straightforward (e.g., increasing the problem scale for TSP). Is there any general or systematic way of scaling the complexity of optimization problem?
* The problem scales considered in this paper are too small. For example, in the case of TSP, only instances with fewer than 20 cities are evaluated, which is far from the scale encountered in real-world problems.
* Although extensive experiments are conducted, the conclusions (e.g., those derived in Section 4.2) are largely intuitive and unsurprising. However, the analysis in Section 4.3 is interesting. More in-depth analysis (e.g., examining why LLMs (PTR) perform well on mathematical tasks but struggle with complex NP-hard problems) would be particularly interesting.

----
Overall, I believe this paper would be a good work after addressing identified issues through a major revision.

---

> ### Author Rebuttal · Authors · 2026-03-30
>
> **Weakness 1**
>
> We thank the reviewer for noting this issue. We have already synced with the AC and clarified the source of the formatting problem.
>
> ---
>
> **Weakness 2 and 4:  Scale Comparison: OPT-Engine vs. Existing Benchmarks**
>
> OPT-Engine is an extensible framework that generates instances with controllable complexity, enabling systematic evaluation of performance as problem scale increases.
>
> The following table compares OPT-Engine with existing benchmarks:
>
> |Dataset|Type|#Inst|Generation|TSP|Netflow|
> |-|-|-|-|-|-|
> |IndustryOR|Static|100|Human|7 n|11n|
> |OptMATH|Static|166|Syn+Human|7n|8n|
> |OptiBench|Static|605|Syn+Human|/|/|
> |OPT-Engine|Extensible|2,690+|Solver-verified|20+n|20n|
>
> We acknowledge that 20-node TSP instances do not reflect real-world complexity.
> To partially mitigate this limitation, we extended our analysis to 45 nodes, enabling a more robust scaling comparison between TIR and PTR.
>
> | Cities | PTR | TIR |
> |-|-|-|
> | 5      | 0.80| 0.80|
> | 10     | 0.20| 0.90|
> | 15     | 0.00| 0.90|
> | 20     | 0.00| 0.90|
> | 25     | 0.00| 1.00|
> | 30     | 0.00| 0.80|
> | 35     | 0.00| 0.70|
> | 40     | 0.00| 0.80|
> | 45     | 0.00| 0.90|
>
> As shown, the results are consistent with the main findings: TIR remains robust, while PTR collapses to 0% beyond 15 cities.
>
> ---
>
> **Weakness 3: Other approaches for Complexity Scaling**
>
> We agree that complexity scaling is inherently problem-family dependent: TSP, netflow and so on do not share a single natural scale parameter. Within each class, however, our scaling is systematic, varying the core structural size in a controlled way while keeping the rest of the generation pipeline fixed. We will clarify this in the revision and note that a more unified notion of complexity scaling remains an important future direction.
>
> ---
>
> **Weakness 5: Limitation of PTR for optimization modelings**
>
> We thank the reviewer for this insightful inquiry on inherent limitations of PTR in optimization modeling. In response, we have expanded our analysis to quantify the Feasibility Rate (F), Accuracy Rate (A) across both paradigms.
>
> |  | Paradigm | Easy (F/A) | Medium (F/A) | Hard (F/A) |
> | --- | --- | --- | --- | --- |
> | TSP | PTR | 97.1% / 79.1% | 92.0% / 11.0% | 80.0% / 4.0% |
> | | TIR | 88.3% / 88.3% | 84.3% / 84.3% | 82.0% / 82.0% |
> | Knapsack | PTR | 86.3% / 82.5% | 63.3% / 63.3% | 57.1% / 47.1% |
> | | TIR | 96.3% / 96.3% | 91.7% / 91.7% | 95.7% / 95.7% |
> | Netflow | PTR | 70.0% / 10.0% | 60.0% / 0.0% | 55.0% / 0.0% |
> | | TIR | 100.0% / 100.0% | 100.0% / 100.0% | 100.0% / 100.0% |
>
> These results clarify the distinct failure modes of two paradigms.
>
> -For **TIR**, accuracy is tied to feasibility: once the model captures correct logical structures, the solver reliably handles the optimization. So failures mainly stem from mis-formulating constraints or objectives.
>
> -For **PTR**, two limitations emerge: **reasoning collapse**, where feasibility drops sharply as difficulty increases because the model struggles to optimize in-context while preserving global mathematical structure, and **calculation limitations** create a large feasibility–accuracy gap, so PTR can produce plausible answers yet still miss the exact optimum (e.g., medium-scale TSP: 92.0% feasible vs. 11.0% accurate).
>
> ---
>
> **Q-1: Taxonomy of code‑evolution in LLM‑driven optimization**
>
> We thank the reviewer for raising this line of work.
> The key distinction is that **code evolution** improves heuristic/solver code for structured instances (e.g., `.mps`, CVRPLIB, TSPLib), while **NL-to-Opt modeling** (our setting) maps natural-language optimization problems into formulation and solver code.
>
> The following table summarizes key distinctions:
>
> |Aspect|Code Evolution|NL-to-Opt Modeling|
> |-|-|-|
> |**Input Source**|Problem instances (.mps) and code|NL problem descriptions|
> |**Output Format**|Improved heuristic code |Formulation and solver code|
> |**Primary Metric**|Solution quality (Gap%), time|Formulation accuracy, mapping|
>
> References:
>
> [1] Liu et al., "Evolution of Heuristics: Towards Efficient Automatic Algorithm Design Using Large Language Model," ICML 2024.
>
> ---
>
> **Q-2: Example of objective perturbations and constraint augmentation**
>
> We will add concrete examples to the Appendix. Here, we provide one for TSP, where we highlight the key parts for a clear view:
>
> |Type|Template|
> |-|-|
> | Objective  | ... The distances between cities are measured in kilometers:  A to B: 184.2 km, A to C: 71.6 km, ..., C to D: 64.0 km. **In addition, due to mandatory detours, the driver must travel an extra fixed distance of 10 kilometers in total no matter what the route is**. The goal is ...|
> | Constraints | ... The distances between cities are measured in kilometers: A to C: 71.6 km, A to D: 94.6 km, ..., C to D: 64.0 km. There is no direct road between City A and City B. **In addition, exactly one of the following two roads must be included in the tour: the road between City B and City C, the road between City C and City D.** The goal is ... |

---

> > ### Author Rebuttal · Reviewer_1o4E · 2026-04-02
> >
> > Thanks for your rebuttal. Please see my comments below:
> > * W2: Thanks for your summary. I would like to see a conceptual comparison with recent benchmarks.
> > * W3: Beyond problem size, complexity scaling could also be examined along the problem distribution axis, or through their interaction. For instance, a large problem with a simple distribution may be less challenging than a smaller problem with a more complex distribution for deep learning–based methods. Does a similar pattern hold for LLM-based approaches?
> > * W4: A scale of 45 nodes remains relatively small. Recent LLM-based approaches typically handle instances with hundreds of nodes.
> > * Q1: Have you considered using LLMs to translate natural language into structured instances, and then solving them via code-evolution methods? I raise this because PTR appears intuitively less promising, particularly for larger instances given the NP-hard nature of the problems.
> >
> > ----
> >
> > Thanks for your further response to my questions. I will increase the score accordingly.

---

> > > ### Author Response · Authors · 2026-04-02
> > >
> > > ---
> > > **W2:  Comparison with Recent Benchmark**
> > >
> > > We note that there is another recent work in the community addressing the issue that existing benchmarks are too small in scale: Constructing Industrial-Scale Optimization Modeling Benchmark (MIPLIB-NL, Feb 10, 2026). These two works represent fundamentally different methodological approaches and research directions:
> > >
> > >    1. **Construction approaches**: MIPLIB-NL utilizes a reverse translation pipeline: it takes existing optimization instances from MIPLIB (within the optimization community) and translates them back into natural language (NL) problems. In contrast, OPT-Engine employs a forward construction pipeline, relying on expert-designed algorithmic generators that produce extensible numeric instances and then extend them to NL descriptions.
> > >
> > >    2. **Extensible vs. Static** : As MIPLIB-NL relies on existing instances, it provides fixed, massive-scale problems. OPT-Engine's forward generation, by contrast, allows for parameterized, continuous scaling (e.g., scaling TSP from $N=5$ to $N=100$).
> > >
> > >    3. **Diagnostic Objective**: While MIPLIB-NL establishes a new high-water benchmark for the community, OPT-Engine serves as a diagnostic tool. Its controllable scaling uniquely enables systematic evaluation of different reasoning paradigms.
> > >
> > > | Feature | **OPT-Engine** | **MIPLIB-NL** |
> > > | :- | :- | :- |
> > > | **Construction** | Forward Construction (Expert Generators) | Reverse Translation (instance to NL) |
> > > | **Instance Source** | Expert-design  code | Historical MIPLIB database |
> > > | **Scaling** | **Controllable & Granular** | Static (Fixed by original instance) |
> > > | **Goal** | Diagnosing Framework |  End-to-End  benchmark |
> > >
> > > ---
> > > **W3:  Other approaches for Complexity/Difficulty**
> > >
> > > Beyond raw problem scale, the intrinsic complexity of an optimization task is governed by its structural topology, particularly the constraint-to-variable ratio. In Section 5, we identify three orthogonal axes of difficulty: Linguistic Complexity, Objective Perturbation, and Constraint Augmentation. To illustrate the impact of structural density, we provide a comparative analysis of accuracy with and without constraint augmentation:
> > >
> > >  Paradigm | Easy | Medium | Hard |
> > > |:-|-:|-:|-:|
> > > | TIR | 88.3% | 84.3% | 82.0% |
> > > | TIR(+c) | 55.0% | 76.0% | 68.0% |
> > > | PTR | 79.1% | 11.0% | 4.0% |
> > > | PTR(+c) | 15.0% | 6.0% | 0.0% |
> > >
> > >
> > > ---
> > > **W4:   small-scale TSP instances**
> > >
> > >  We clarify that OPT-Engine is an extensible framework . To address your's concern, we extended the TSP experiments to 110+ nodes.
> > > | Cities | PTR | TIR |
> > > |-|-|-|
> > > | 10     | 0.2| 0.9|
> > > | 20     | 0.0| 0.9|
> > > | 30     | 0.0| 0.8|
> > > | 40     | 0.0| 0.8|
> > > | 50     | 0.0| 0.8|
> > > | 60     | 0.0| 0.7|
> > > | 70     | 0.0| 0.8|
> > > | 80     | 0.0| 0.6|
> > > | 90     | 0.0| 0.6|
> > > | 100     | 0.0| 0.5|
> > > | 110     | 0.0| 0.6|
> > >
> > > Since the primary objective of this paper is to benchmark the fundamental limits of mainstream reasoning paradigms, we believe that scaling the TSP problem to 110 nodes already demonstrates the performance ceiling of different paradigms.
> > >
> > > ---
> > >
> > > **Q-1: NL problem to instances construction for code-evolution**
> > >
> > > Thank you for your thoughtful question.
> > > You suggest using LLMs to translate NL problems into structured instances and then solving via code‑evolution methods.
> > > That's a very interesting direction. We would like to gently share a few thoughts for your consideration:
> > >
> > > 1. **Modeling bottleneck remains the core challenge** :
> > > In the NL4Opt setting, if we translate the NL inputs into a correct mathematical formulation and solver code (e.g., Gurobi), we can trivially output structured instances via "model.write("instance.mps")". However, the high-fidelity NL-to-formulation and code translation remains essential. This is exactly the "modeling bottleneck" our benchmark evaluates.
> > >
> > > 2. **Code‑evolution is task‑specific**:
> > > Current code-evolution methods excel when applied to fixed problem classes that share the same underlying structure (e.g., evolving TSP heuristics on TSPLIB or CVRP heuristics on CVRPLIB). Due to the differences between problem types, an improved TSP heuristic cannot be directly extended to solve CVRP instances. In contrast, the current NL4Opt benchmark requires models to tackle a diverse, cross-domain set of  NL problems. Extending code evolution to such a broad and heterogeneous problem set would require intensive, case-by-case adaptation, which limits its generalizability.
> > >
> > > 3. **A hybrid approaches** :
> > > We agree that the boundaries between these methodologies are becoming blurred. A promising extension would be a hybrid pipeline: first, use an LLM to generate an initial heuristic based on the NL description and partial instance structure; then, apply code evolution to refine that heuristic. However, applying heuristic evolution to a library like MIPLIB—which contains highly diverse problem structures, remains a significant challenge.
> > >
> > > ---
> > > We sincerely appreciate the reviewer's feedback and hope that we have addressed all of the comments above.

---

### Official Review · Reviewer_p7Fn · 2026-03-14

**Soundness:** 3
**Presentation:** 2
**Significance:** 2
**Originality:** 2
**Overall Recommendation:** 3
**Confidence:** 3

**Summary:**

This paper presented a new evaluation framework for optimization problems. The benchmark contains 10 problems and each can be sampled with random setups.

This paper also have extensive experiments and analysis with different reasoning paradigms (Tool-Integrated-Reasoning and Pure-Text-Reasoning).

**Compliance With Llm Reviewing Policy:**

Affirmed.

**Final Justification:**

In general I agree this work is complete and valid. Most of my concerns are addressed. The main remaining concerns is still the construction of dataset. Naively sampling optimization problem might not lead to representative conclusions.  However, as LLM is not good at optimization task (task not saturated yet), this might not be obvious yet.

As a conclusion, I agree with weak accept.

**Key Questions For Authors:**

- In the second part of the Section 4.2, the figure 4. It's not very fair to compare -instruct and ptr with -rl model, as the rl model is trained with in-distribution data. Figure 4 can only tell the improvement by training, and the paper should clarify the details of data used to train (what complexity level, etc.)

- There should be a results table for all question types shown in Figure 2, to show how the latest LLMs perform on the presented problems.

**Limitations:**

The paper has Impact statement, but no Limitations.

The paper should discuss the limitations of the current sampling mechanism. What's the setup coverage?

**Strengths And Weaknesses:**

Strengths:

- The optimization task addressed is interesting and of good difficulty for the latest LLMs.

- The present framework is not static, which is relatively robust.

- The analysis is generally good. Especially the comparison of TIR and PTR methods, and how the answer length increases as the question complexity increases.


Weakness:

- The included problems are standard optimization problems, which have limited novelty in general. One critical drawback is that number sampling is random, which makes the long-tailed setup very hard to be sampled.

- It would be better to have a dedicated comparison table to show the difference between the proposed dataset and previous datasets, including the dimensions of scale, difficulty and so on.

---

> ### Author Rebuttal · Authors · 2026-03-31
>
> **Weakness 1.1 & Weakness 2:  Benchmark Novelty and Scale**
>
> We thank the reviewer for the opportunity to clarify our contribution.
> While these problem classes are canonical, OPT-Engine’s novelty lies in its paradigm shift from a static dataset to a dynamic, extensible benchmark engine. It provides a standardized pipeline for systematic analysis of LLM performance under complexity scaling (e.g., N=5 to N=45 for TSP), whereas existing static benchmarks are restricted to small-scale instances lacking complexity to "stress-test" modern LLMs.
>
> The following table provides a comparison:
>
> |Dataset|Type|#Inst|Generation|TSP|Netflow|
> |-|-|-|-|-|-|
> |IndustryOR|Static|100|Human|7n|11n|
> |OptMATH|Static|166|Syn+Human|7n|8n|
> |OptiBench|Static|605|Syn+Human|/|/|
> |OPT-Engine|Extensible|2,690+|Solver-verified|45+n|20n|
>
> **Rigorous Verification** :
> During pipeline design, we manually curated expert promp expert prompt templates and instance-construction code to ensure semantic fidelity and mathematical solvability for every problem. Precisely, this rigor makes these canonical problems a sound, reproducible foundation for benchmarking.
>
> ---
> **Weakness 1.2 : Diversity for Sampling mechanism**
>
> We appreciate your valuable suggestion to move beyond uniform sampling.
> To examine this concern, we extended OPT-Engine to support non-uniform sampling mechanisms, including Gaussian, Poisson, and Exponential. As a concrete example, we generated TSP instances under a Poisson-style long-tail setup, where 80% of cities lie in a dense core while 20% are distant outliers, to simulate real-world spatial skew.
>
> The full results are summarized below:
>
> | #Cities | PTR (Uni) | PTR (Pois) | TIR (U) | TIR (Pois) |
> |-|-|-|-|-|
> |6|84%|30%|70%|60%|
> |8|50%|20%|100%|70%|
> |10|30%|0%|73%|90%|
> |12|10%| 0%|80%|80%|
> |14| 5%| 0%|100%| 100%|
> |16| 5%|0%|70%| 70%|
> |18|5%|0%|100%|80%|
> |20|0%|0%|100%|90%|
>
> As shown, under both sampling schemes, PTR degrades rapidly as problem size increases, while TIR remains consistently robust. Notably, the sharper drop of PTR under Poisson sampling suggests that skewed structures introduce additional difficulty to PTR.
>
> ---
> **Q-1: Clarification on RL Training Data and Comparison Fairness**
>
>
> We thank the reviewer for the critical observation. Below, we address three key aspects:
>
>    1. **Execution Failures**: TIR-RL mitigates solver execution errors for Qwen3-4B. By decoupling tool-calling syntax from mathematical reasoning, it ensures TIR results reflect true modeling capacity rather than execution failures.
>
>    2. **OOD Evaluation**:  The training set comprises 10,000 samples from SIRL and OptMath repositories. While TSP training instances use $n \in [5, 10]$, our evaluation set extends to $n = 20$, representing an Out-of-Distribution (OOD) shift in problem scale.
>
>   3. **Controlled Comparison**: To ensure a fair comparison, we retrained PTR-RL using the same data and configuration as TIR-RL.
>
> Below are comparative performances across different Qwen3-4B variants:
>
> | Problem Class | Paradigm | Easy (%) | Medium (%) | Hard (%) |
> | - | - | - | - | - |
> | Knapsack | TIR | 17.5 | 16.7 | 32.9 |
> | | RL-TIR | 96.3 | 91.7 | 95.7 |
> | | PTR | 50.0 | 6.7 | 5.7 |
> | | **RL-PTR** | 86.4 | 55.0 | 18.3 |
> | Netflow | TIR | 40.0 | 40.0 | 60.0 |
> | | RL-TIR | 100.0 | 95.0 | 95.0 |
> | | PTR | 10.0 | 0.0 | 0.0 |
> | | **RL-PTR** | 10.0 | 10.0 | 0.0 |
> | Portfolio | TIR | 42.0 | 26.0 | 33.3 |
> | | RL-TIR | 100.0 | 98.0 | 90.0 |
> | | PTR | 18.0 | 4.0 | 1.7 |
> | | **RL-PTR** | 36.7 | 12.0 | 4.0 |
>
> As shown, while PTR-RL achieves high accuracy, its performance decays as problem complexity increases. This confirms the "Modeling Bottleneck" for PTR persists even with reasoning capability enhanced by RL.
>
> We commit to **releasing all training recipes, dataset specifications, and 10k samples** for total transparency.
>
> ---
> **Q-2: Performance across SOTA Models**
>
> We thank the reviewer for the suggestion. In the table below, we evaluate both PTR and TIR across four problem classes using recent SOTA models: DeepSeek‑V3.2, GPT‑5.1, and Gemini‑3.0‑Pro.
>
> | Problem | Paradigm | DS (%) | GPT (%) | Gemini (%) |
> |-|-|-|-|-|
> | Knapsack | TIR | 94.8 | 95.2 | 93.9 |
> | | PTR | 65.2 | 66.5 | 93.1 |
> | Production | TIR | 95.0 | 82.5 | 86.3 |
> | | PTR | 15.0 | 12.5 | 61.3 |
> | Portfolio | TIR | 90.0 | 90.0 | 90.0 |
> | | PTR | 21.9 | 35.6 | 79.4 |
> | NetFlow | TIR | 100.0 | 100.0 | 100.0 |
> | | PTR | 2.0 | 4.0 | 96.0 |
>
>
> **Limitations**
>
> We thank the reviewer for raising this point. We will move the limitation discussion in **Section 6** to a more prominent position and explicitly detail **sampling mechanism**. In the current benchmark, coverage is **problem-specific** and based on uniform sampling: e.g., **TSP** (4–20 cities, coords $\in [0,200]^2$), while **Netflow** (3–15 nodes, supplies $\in [10,100]$, shipping costs $\in [1,10]$, and capacities $\in [5,100]$). We will add discussions of such sampling effects and distributional diversity to our limitations discussion.

---

> > ### Author Rebuttal · Reviewer_p7Fn · 2026-04-05
> >
> > The contribution of dynamic benchmark is still limited. There are plenty of existing benchmarks have dynamic elements in their design. Maybe not much in OR domain, but quite many in math, code and reasoning domains. Therefore in general I would agree its novelty. The diversity of sampling need to be more throughly discussed and experimented to show the validity and effectiveness.
> >
> > The rest of the questions are addressed well. Overall I would maintain the original score.

---

> > > ### Author Response · Authors · 2026-04-05
> > >
> > > We thank the reviewer for the follow-up.
> > >
> > > As we have already addressed the sampling diversity question in our previous rebuttal--we provided an additional ablation study comparing uniform vs. long-tail sampling and reported evaluation metrics through problem size scaling.
> > >
> > > We now turn to the reviewer’s second concern: why dynamic benchmarking is needed in OR or optimization modeling domain.
> > >
> > > ---
> > > While current LLMs (mainly via chain‑of‑thought reasoning paradigm) have achieved super‑human performance on difficult benchmarks(AIME), their performance in optimization modeling, translating natural language into mathematical optimization models, remains limited.
> > >  Our paper attacks this gap by revealing two issues:
> > >
> > >    1. **Identifying failure modes of Chain‑of‑Thought (CoT, referred to as PTR in our paper)**:
> > > We show that while chain-of-thought (CoT) reasoning is effective for mathematical problem solving, it fails in optimization tasks. The core issue is that LLMs cannot maintain global constraint consistency across reasoning steps. Even when numeric accuracy is achieved (e.g., correct arithmetic), the model frequently violates feasibility constraints—such as exceeding capacity or breaking precedence. This failure mode is unique and not captured by existing math benchmarks.
> > >
> > >    2. **Bottleneck of TIR (Tool‑Integrated Reasoning, current SOTA)**
> > >     Further, we identify why TIR's performance remains suboptimal: although it can utilize external solvers to offload computational burden, it still struggles to generate correct model formulations. We isolate the key variations—constraint density, objective function types, and the deconstruction of these factors—that contribute to this failure. The bottleneck lies mainly in the constraint matrix—a challenge that only a dynamic OR benchmark can expose.
> > >
> > > ---
> > > We  thank the reviewer for your time and the effort invested in providing this feedback,
> > >  and hope that we have addressed all of the comments above.

---

### Official Review · Reviewer_3PsW · 2026-03-24

**Soundness:** 2
**Presentation:** 3
**Significance:** 3
**Originality:** 3
**Overall Recommendation:** 4
**Confidence:** 5

**Summary:**

This paper studies how well current LLM-based methods solve optimization problems as problem complexity increases and the problem structure shifts out of distribution. It shows that tool-integrated reasoning is crucial for maintaining performance under higher complexity, while most existing methods remain highly sensitive to changes in constraint formulations, which emerges as a major bottleneck in robust optimization modeling.

**Compliance With Llm Reviewing Policy:**

Affirmed.

**Final Justification:**

My concerns have been well addressed. I remain positive about this paper and raise my confidence on the assessment.

**Key Questions For Authors:**

See weaknesses.

**Limitations:**

yes

**Strengths And Weaknesses:**

**Strengths**:
1. The paper studies an interesting and meaningful question: how current LLM-based optimization methods behave as problem complexity increases and as problem formulations shift away from canonical distributions. This is an important evaluation setting for understanding the capability boundaries of these methods.
2. I appreciate the paper's diagnosis that sensitivity to constraint variations is a central failure mode. This finding is both interesting and valuable for future work on robust optimization modeling with LLMs.


**Weaknesses**:
1. The evaluation relies heavily on binary accuracy under an exact-solution paradigm, which may be too coarse for NP-hard problems such as TSP. For larger instances, exact optimality is often computationally unrealistic and also not the main practical objective. As the paper itself notes, near-optimal solutions found by heuristic-style reasoning can still be useful in practice. I therefore think it would strengthen the evaluation to additionally report continuous quality metrics, such as optimality gap, feasibility rate, and runtime, especially as problem complexity increases.
2. I agree that the paper identifies sensitivity to constraint variations as an important failure mode. However, I am not fully convinced that this diagnosis is specific to constraints, rather than reflecting a broader weakness of LLMs on optimization problems that deviate from familiar training problem distributions. In particular, the current objective perturbation is only a constant shift, which preserves both the feasible set and the ordering of solutions, and therefore does not meaningfully alter the mathematical structure of the problem. Because of this, the contrast between “objective perturbation” and “constraint augmentation” may currently be too weak to support a strong conclusion that constraints are uniquely the primary bottleneck. It would be more convincing to include objective-side perturbations that genuinely change the optimization structure.

---

> ### Author Rebuttal · Authors · 2026-03-29
>
> We sincerely thank the reviewer for the thoughtful and constructive feedback. Your insights on evaluation granularity and objective perturbations have significantly strengthened both our experimental protocol and the framing of our paper.
>
> **Weakness 1**:
>
> We agree that exact optimality should be complemented by more graded metrics, as near-optimal solutions are often practically useful for harder instances.
>
> In response, we conducted a comparative analysis between  TIR and PTR across five MILP classes using DeepSeek-V3.2, stratifying instances into **Easy, Medium, and Hard**. Beyond exact accuracy, we now report the Optimality Gap and a Feasible-Answer Rate.
>
> **Table 1: Feasible-Answer Rate (F), Accuracy (A), and Optimality Gap (Gap)**
>
> | Problem | Paradigm | Easy | Easy | Medium | Medium | Hard | Hard |
> |---|---|---|---|---|---|---|---|
> | | | F/A | Gap | F/A | Gap | F/A | Gap |
> | TSP | PTR | 97.1%/79.1% | 0.91% | 92.0%/11.0% | 8.47% | 80.0%/4.0% | 13.12% |
> | | TIR | 88.3%/88.3% | 0.00% | 84.3%/84.3% | 0.01% | 82.0%/82.0% | 0.00% |
> | Knapsack | PTR | 86.3%/82.5% | 0.19% | 63.3%/63.3% | 0.00% | 57.1%/47.1% | 0.42% |
> | | TIR | 96.3%/96.3% | 0.00% | 91.7%/91.7% | 0.00% | 95.7%/95.7% | 0.00% |
> | Binpacking | PTR | 91.1%/90.0% | 0.12% | 66.4%/64.6% | 0.13% | 42.7%/32.7% | 0.71% |
> | | TIR | 85.6%/85.6% | 0.00% | 68.2%/68.2% | 0.00% | 67.3%/67.3% | 0.00% |
> | Jobshop | PTR | 70.0%/40.0% | 2.73% | 16.7%/3.3% | 2.22% | 1.7%/0.0% | 0.04% |
> | | TIR | 95.0%/95.0% | 0.00% | 90.0%/90.0% | 0.00% | 56.7%/56.7% | 0.00% |
> | Netflow | PTR | 70.0%/10.0% | 6.01% | 60.0%/0.0% | 8.92% | 55.0%/0.0% | 12.42% |
> | | TIR | 100.0%/100.0% | 0.00% | 100.0%/100.0% | 0.00% | 100.0%/100.0% | 0.00% |
>
> Here, *exact accuracy* follows the definition in the paper.
> The *optimality gap* is calculated as $\frac{|\hat{y}-y^\star|}{|y^\star|}$, where $\hat{y}$ and $y^*$ are output and true optimal values respectively. As PTR outputs only a final answer rather than a full solution vector, we utilize an **answer-level feasibility rate** to distinguish near-valid outputs from clearly invalid ones. An output is counted as feasible if it strictly satisfies:
> 1. Runtime Limit (TIR only): Solver execution is within 100s per instance.
> 2. Directional Validity: The output value respects the optimization direction.
> 3. Deviation Bound: $|\hat{y}-y^\star| \le 0.6|y^\star|$.
>
> These metrics reveal distinct failure modes:
>    1. **Under TIR**, accuracy is tied to feasibility: once the model captures correct logical structures, the solver reliably handles the optimization. So failures mainly stem from mis-formulating constraints or objectives.
>    2. **Under PTR**, there are two distinct failure modes:
>       * **reasoning collapsep**: as complexity increases, feasibility drops sharply as difficulty increases, because the model struggles to optimize in-context while preserving global mathematical structure.
>       * **calculation limitations**: create a large feasibility–accuracy gap, so PTR can produce directionally plausible answers yet still miss the exact optimum (e.g., medium-scale TSP: 92.0% feasible vs. 11.0% accurate).
>
> **Weakness 2**:
>
> We fully agree with the reviewer's insightful observation that a constant objective shift $f(x)+C$ merely translates the value without altering the fundamental mathematical structure.
>
> To provide a more rigorous test, we applied non-isomorphic perturbations to improve experiments. If the original objective is $f(x)=\sum_i a_i x_i+b$, we perturb it to **$\sum_i a'_i x_i + b$**. Modifying variable coefficients reshapes the optimization landscape, guaranteeing a shift in the optimal solution.
> E.g.,  an additional constraint in TSP could be : **"Due to road damage, the one-way driving distance from City A to City B is doubled, whereas the return trip from City B to City A remains unaffected."** This provides a conceptually simple but effective mechanism to test the altered optimal route.
>
> **Table 2: TSP Accuracy under Structural Objective Perturbations**
>
> | Paradigm | Perturbation | Easy | Medium | Hard |
> |---|---|---|---|---|
> | TIR | Original | 88.3% | 84.3% | 82.0% |
> | | New Obj. | 90.0% | 90.0% | 74.0% |
> | PTR | Original | 79.1% | 11.0% | 4.0% |
> | | New Obj. | 73.3% | 28.0% | 6.0% |
>
> Results show that even after introducing structure-changing objective perturbations, the overall performance trend remains stable for both paradigms. In particular, the impact of objective-side changes is still notably smaller than that of constraint-side shifts studied in our paper. We thank the reviewer for this helpful suggestion, which prompted us to refine both experiments and the wording of our claim. In the revised manuscript, we will improve our experiments and clarify that **constraint formulation appears to be the most severe bottleneck among the perturbation types we evaluated**, while avoiding any broader conclusion beyond the empirical scope of our experiments.

---

> > ### Author Rebuttal · Reviewer_3PsW · 2026-04-03
> >
> > Thank you for the rebuttal. I think my main concerns have been adequately addressed. In particular, the paper identifies constraint handling as an interesting failure mode of current methods, which I believe is a valuable finding for the future development of the field. Overall, I remain positive about the paper.

---

> > > ### Author Response · Authors · 2026-04-04
> > >
> > > Thank you for your final comments and for your constructive engagement throughout this review process.
> > >
> > > We are especially grateful for your feedback regarding the feasibility rate analysis of the **PTR (Classical Chain-of-Thought approaches for mathematical reasoning)**. Your suggestion prompted additional empirical experiments that clarified a crucial point: classical Chain-of-Thought  often fails in optimization modeling also because it struggles to capture the global optimization conditions inherent in such tasks.
> > >
> > > This insight has significantly strengthened our manuscript. We will carefully review your comments and will integrate them into  the final version.
> > >
> > > Your guidance has been tremendously helpful.

---

### Decision · Program_Chairs · 2026-04-30

**Decision:**

Accept (regular)

**Comment:**

After rebuttal, all the reviewers indicate that the major concerns have been addressed, although one of the reviewers did not update the score. Some minor concerns remained there include 1) construction of the dataset, 2) the PTR setting might not be promising for complex optimization problems; 3) benchmark is (only) conducted on small-scale instances; 4) lack of a systematic way for complexity scaling. I recommend 'weak accept', as I believe it will offer some useful insight for researchers of this topic, but I also suggest the authors to explicitly discuss/address the above remaining concerns in the final version.